# SMELLNET: A LARGE-SCALE DATASET FOR REAL-WORLD SMELL RECOGNITION

**Dewei Feng, Wei Dai, Carol Li, Alistair Pernigo, Paul Pu Liang**
MIT Media Lab and MIT EECS
`https://github.com/MIT-MI/SmellNet`

## ABSTRACT

The ability of AI to sense and identify various substances based on their smell alone can have profound impacts on allergen detection (e.g. smelling gluten or peanuts in a cake), monitoring the manufacturing process, and sensing hormones that indicate emotional states, stress levels, and diseases. Despite these broad impacts, there are few standardized datasets, and therefore little progress, for training and evaluating AI systems' ability to 'smell' in the real-world. In this paper, we use small gas and chemical sensors to create SMELLNET, a comparatively large dataset for sensor-based machine olfaction that digitizes a diverse range of smells in the natural world. SMELLNET contains about 828,000 time-series data points across 50 substances, spanning nuts, spices, herbs, fruits, and vegetables, and 43 mixtures among them with fixed ingredient volumetric ratios, with 68 hours of data collected. Using SMELLNET, we developed SCENTFORMER, a Transformer-based architecture combining temporal differencing and sliding-window augmentation for smell data. For the SMELLNET-BASE classification tasks, SCENTFORMER achieves 63.3% Top-1 accuracy with GC-MS supervision, and for the SMELLNET-MIXTURE distribution prediction tasks, SCENTFORMER achieves 50.2% Top-1@0.1 on the test-seen split. SCENTFORMER's ability to generalize across conditions and capture transient chemical dynamics demonstrates the promise of temporal modeling in sensor-based olfactory AI. SMELLNET and SCENTFORMER lay the groundwork for sensor-based olfactory applications across healthcare, food and beverage, environmental monitoring, manufacturing, and entertainment.

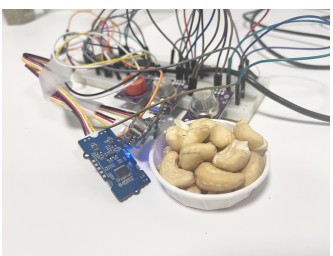
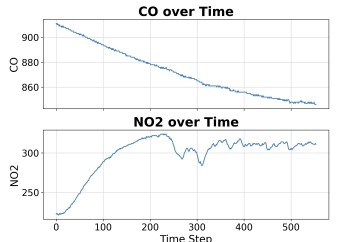
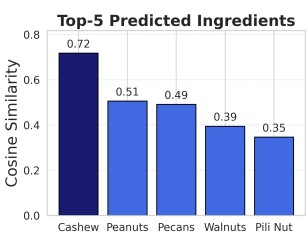

(a) Sensor setup detecting cashew.    (b) Time-series signals from CO and $NO_2$ sensors.    (c) Top-5 model predictions using cosine similarity.

Figure 1: **Overview of our smell sensing data collection and modeling pipeline.** (a) Sensor hardware setup and data capture. (b) Raw sensor readings over time. (c) AI model predictions on the substance.

## 1 INTRODUCTION

Advancements in AI have revolutionized how machines perceive and interact with the world. However, most progress has been limited to the text, vision, and audio modalities (Liang et al., 2024; Gan et al., 2022). The human sense of smell is crucial for environmental perception, social interaction, and regulating well-being. Similarly, sensor-based smell recognition (machine olfaction) can support a range of applications in the entertainment, e-commerce, manufacturing, and food and beverage industries (Deshmukh et al., 2015; Vilela et al., 2019). More ambitiously, smell-based diagnostics can help in early disease detection (e.g., COVID-19) (Kwiatkowski et al., 2022), and even 'smelling' hormones and indicators of emotional states, stress levels, and for early prognosis of cancer (Tillotson, 2017; Zamkah et al., 2020).

Table 1: **Representative olfactory datasets by modality.** Human-perception datasets pair odorants with human judgments, and sensor datasets pair odor sources with instrument signals. SMELLNET is the most diverse portable metal-oxide (MOX) sensor time-series dataset in this table, covering natural foods and controlled mixtures, with ingredient-level GC-MS priors.

| Dataset | Stimulus | What's measured | Acquisition / method | Scale (rep.) |
|---|---|---|---|---|
| **Human-perception datasets (odorant/mixture + human response)** | | | | |
| Dravnieks Atlas (Dravnieks, 1985) | Mono-molecular | Semantic descriptor ratings | Human evaluation | 21.3k ratings |
| DREAM Challenge (Keller et al., 2017) | Mono-molecular | Semantic descriptor ratings | Human evaluation | 10.0k ratings |
| Olfactory Metamers (Ravia et al., 2020) | Molecular mixtures | Perceptual similarity judgments | Human evaluation | 49.8k judgments |
| **Sensor / machine-olfaction datasets (odor source + sensor signals)** | | | | |
| Coffee quality e-nose (Rodríguez et al., 2010) | Coffee samples | Gas-sensor time series | E-nose (8 gas sensors), 1 Hz for 300 s | 58 measurements |
| Beef quality e-nose (Wijaya et al., 2018) | Beef cuts | Gas-sensor time series (+ env in dataset) | E-nose (MQ sensors) in uncontrolled ambient conditions | 5 sessions (2160 min each) |
| Gas Sensor Array Drift (Vergara, 2012) | Pure gases | MOX array responses / features | 16 MOX sensors, drift over 36 months | 13,910 measurements |
| **SMELLNET (ours)** | Natural foods and ingredient mixtures | Base: 6 channels, Mixture: 4 channels (Grove only) | Low-cost MOX array, controlled acquisition protocol | 50 substances, 43 mixtures, 68 h, 828k timesteps |

Nevertheless, machine olfaction with deployable sensors remains far less developed when compared to computer vision and natural language processing. A key bottleneck is the lack of standardized, realistic sensor-side benchmarks for learning and evaluating models on temporal smell signals. We believe that larger-scale data and real-time AI models can enable richer feature representations of smell for accurate sensing, classification, and feature fusion between smell and other human senses. This strategy differs from past research, which has emphasized feature engineering, small datasets, and simple classification models (Achebouche et al., 2022; Guerrini et al., 2017; Lee et al., 2012), and processing pre-recorded smell data collected via large chemistry lab equipment (Tran et al., 2019; Lee et al., 2023), which do not work in real-time.

As a step towards real-world and real-time smell sensing, we present SMELLNET, a comparatively large sensor-side dataset of how food, beverages, and natural objects register on low-cost gas and chemical sensors. SMELLNET is collected by applying small sensors to 50 substances (nuts, spices, herbs, fruits, and vegetables) and 43 ingredient-level mixtures with fixed volumetric ratios across 68 hours of data. Importantly, SMELLNET contains sensor time series and metadata rather than human perceptual ratings. With 828,000 time step readings across environmental conditions, SMELLNET provides a benchmark for training and evaluating machine-olfaction models on substance classification from sensor readouts.

Using SMELLNET, we developed SCENTFORMER, a Transformer-based architecture combining temporal differencing and sliding-window augmentation for smell data. For the SMELLNET-BASE classification tasks, SCENTFORMER achieves 63.3% Top-1 accuracy with GC-MS supervision, and for the SMELLNET-MIXTURE distribution prediction tasks, SCENTFORMER achieves 50.2% Top-1@0.1 on the test-seen split. SCENTFORMER's ability to generalize across conditions and capture transient chemical dynamics demonstrates the promise of temporal modeling in sensor-based machine olfaction. SMELLNET, SCENTFORMER, and the source code are released in our GitHub repository (linked above) and supplementary materials to ensure reproducibility and to facilitate new applications in healthcare, food and beverage, environmental monitoring, manufacturing, and entertainment.

## 2 THEORETICAL BASIS AND RELATED WORK

While large-scale machine olfaction with low-cost sensor time series remains relatively underexplored, we are inspired by human smell sensing, the chemistry and biology of smell, and using AI to process small-scale smell data.

**Human smell receptors:** Olfaction, the sense of smell, allows for the detection and discrimination of odors in the environment (Stevenson, 2010). The human nose can detect and discriminate between an estimated trillion different odors, even in minute quantities (Bushdid et al., 2014). This makes the human olfactory system the largest, in terms of the number and diversity of receptors, allowing for the sensing of a vast chemical landscape (Sharma et al., 2019). Human olfactory perception functions through a combinatorial code, where each odorant molecule binds to a specific set of olfactory receptors (ORs) in the nose (Malnic et al., 1999). This binding converts chemical information into electrical signals which are perceived in the brain (Firestein, 2001).

**Smell sensors** for perceiving smell have been developed, including chemical compositions of gases based on the principles of molecular interaction and chemical potential equilibrium (Brattoli et al., 2011). These sensors can employ different scientific strategies to detect and analyze gas molecules, including those based on semiconducting materials like metal oxides (Nikolic et al., 2020), electro-chemical sensors that generate a current proportional to the gas concentration (Bakker and Telting-Diaz, 2002), optical sensors based on different gases absorbing different wavelengths (Hodgkinson and Tatam, 2012), and conductive polymers that change their conductivity when exposed to gas molecules (Miasik et al., 1986). We use gas sensors due to their portability, although all sensors can suffer challenges due to sensitivity, environmental interference, reproducibility, and device calibration (Sung et al., 2024; Yan et al., 2015).

**AI for smell sensing:** There has been some work in using technology to process pre-recorded smell data, but these systems are not portable or work in real-time. These include graph neural networks trained to classify smell chemical molecules (especially pre-recorded GC-MS data) (Sanchez-Lengeling et al., 2019; Tran et al., 2019; Lee et al., 2023), but they require data collection via large chemistry lab equipment rather than portable sensors. Past research has also emphasized human domain knowledge, feature engineering, small datasets, and simple classification models rather than large-scale data-driven learning (Achebouche et al., 2022; Guerrini et al., 2017; Lee et al., 2012). Electronic noses have been designed to monitor pollutants and for air quality assessment (Attallah and Morsi, 2022; Payette et al., 2023), but they are not generally applicable for any type of smell. Methods to classify biological olfactory data of mice and humans have also been proposed (Fang et al., 2024; Wang et al., 2021), but do not enable portable real-time sensing.

**Other datasets:** Tab. 1 normalizes prior olfaction datasets by the number of data points. Unlike prior work focused on mono-molecular or pairwise human judgments, SMELLNET provides large-scale sensor time series from natural stimuli. To our knowledge, it is the most diverse, open sensor-based dataset for smell, spanning 50 base substances and 43 mixtures over 68 hours.

## 3 CREATING SMELLNET

### 3.1 A SMALL AND REAL-TIME SMELL SENSOR

We use a compact array of low-cost metal-oxide (MOX) gas sensors to capture odor-dependent volatile response patterns. Specifically, our platform uses MQ-3, MQ-5, and the Grove Multichannel Gas Sensor V2, providing six gas-sensor channels corresponding to manufacturer-labeled responses such as carbon monoxide (CO), nitrogen dioxide ($NO_2$), alcohol, volatile organic compounds (VOCs), liquefied petroleum gas (LPG), and ethanol ($C_2H_5OH$). We emphasize that these channel names are manufacturer calibration labels rather than direct measurements of pure analytes in food headspace. In practice, each MOX channel is broadly cross-sensitive to multiple volatile families. This overlapping sensitivity is desirable in an e-nose setting, because the joint multi-channel temporal response provides a structured, discriminative signature of common odors in foods, drinks, and other everyday substances, consistent with prior low-cost e-nose systems used for food-quality monitoring and freshness assessment (Rodríguez et al., 2010; Wijaya et al., 2018; Mahradian, 2023). The circuit diagram and full hardware specifications are provided in App. A.1.

### 3.2 SMELL SENSING DATA COLLECTION

SMELLNET comprises two main components: base substances for classification tasks and mixture substances for distribution prediction tasks.

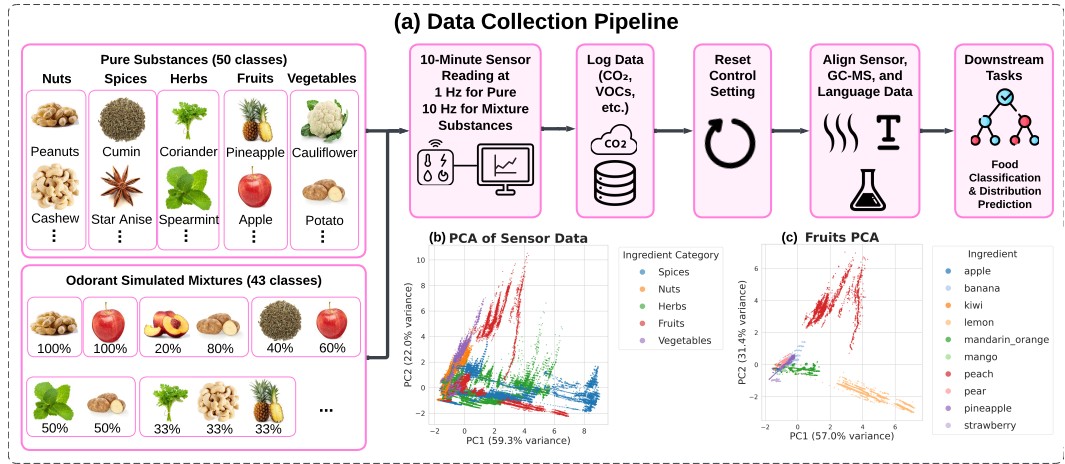

Figure 2: **(a) Data collection pipeline.** Each ingredient undergoes six 10-minute sensing sessions across different days, using a controlled environment to minimize external noise. During each session, 6-channel gas sensor data is recorded at 1 Hz and labeled with the ingredient identity, collection time, and associated metadata. We further pair each ingredient with high-resolution GC-MS data to enable multimodal learning. This setup enables the creation of a structured and temporally rich dataset for representation learning of smells. **(b-c) PCA projections of sensor responses.** While broad category separation is evident, clusters remain partially overlapping, underscoring the challenge of distinguishing ingredients and motivating more advanced models.

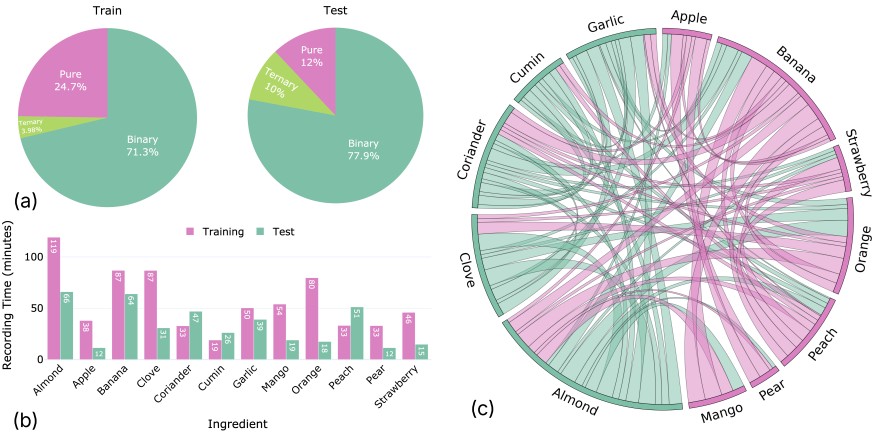

Figure 3: **Composition of SMELLNET-MIXTURE.** (a) Distribution of mixtures across train and test datasets. The test set contains harder samples, including binary (77.9%) and ternary (10%) mixtures. (b) Minutes of valid odor data collected for each ingredient. (c) Ingredient co-occurrence patterns: chords indicate data mixtures, and widths represent the amount of mixture data. Each chord encompasses mixtures of different ratios. Teal chords indicate where the mixture is prominently spices or nuts, while pink indicates mixtures that are prominently fruits on average. The diverse odor mixtures are evenly spread across the entire substance space.

SMELLNET-BASE is collected on 50 substances across 5 classes: nuts, spices, herbs, fruits, and vegetables. The full taxonomy of substances is shown in Fig. 6(b). For each, we used the sensor to collect data for a 10-minute session, repeated 6 times across different days. Each session was done in a controlled environment to minimize environmental factors. At the end of each session, we clear out the air in the controlled environment so that the environment returns to atmospheric conditions (see App. A.2 and App. H.5 for analysis using different time frames and detailed procedures for resetting each experiment). Fig. 2 illustrates the overall data collection pipeline. This gives us 1 hour of sensor readings for each substance and 50 hours of data in total. During each session, 6-channel gas sensor data is recorded at 1 Hz, which gives us 180,000 total data points. Data is labeled with the name of the substance, its detailed description, and the time and date of collection.

In real-world scenarios, mixtures of smells are more common, so we also collect SMELLNET-MIXTURE with mixtures of 12 base odorants. The odorants are selected from various types of fragrance oils, essential oils, and flavor extracts to form our odorant palette: almond, apple, banana,

clove, coriander, cumin, garlic, mango, orange, peach, pear, and strawberry. Details are included in Tab. 17. These odorants span a range of functional groups and chemical families relevant to a broad range of odors, including phenols (clove), aldehydes (almond, cumin), esters (banana, pear, peach, strawberry, apple), terpenes (orange, mango, coriander), and sulfur compounds (garlic). Furthermore, we define two test sets with different generalization challenges: **(1) Test-seen** contains mixture ratios that appear in the training set but from different recording sessions, testing the model's ability to generalize across temporal and environmental variations. **(2) Test-unseen** contains novel mixture combinations never encountered during training, challenging the model's compositional generalization in entirely new ingredient pairings, or novel ratios of familiar mixtures.

Due to size restrictions of the collection environment, the mixture data is collected with only the Grove Multichannel V2 sensor, with four channels spanning across $NO_2$, $C_2H_5OH$, VOC, and CO at 10 Hz. The resulting dataset comprises 18.0 hours of continuous sensor recordings with 648,000 data points across 1,078 distinct measurement sessions, with 679 training sessions (11.32 hours), 215 test-seen mixtures (3.58 hours), and 184 test-unseen mixtures (3.07 hours). Combining the two subsets together, we have a total of 68 hours of sensor readings for 50 base substances and 43 different mixtures, totaling 828,000 data points, making it the largest sensor-based multitask smell dataset to date.

## 3.3 PAIRING WITH GC-MS

One potential limitation of sensor data is its low resolution, which stems from the quality of the sensors used. We therefore paired this data with preexisting open-source pre-recorded GC-MS data (FooDB Contributors, 2024). GC-MS devices are large, bulky, and non-portable, but can detect the exact concentration of various compounds in a substance at a high resolution. As a result, pairing gas sensing data with GC-MS readings provides complementary information for this task and allows for studies of multimodal fusion and cross-modal learning (Liang et al., 2024).

## 3.4 SMELLNET STATISTICS

We standardize the recorded data into a common format via the sensor readings of volatile gases over the time period. We plot some examples of sensor readings in Fig. 2(b-c), and App. B. As shown in Fig. 2(b), we apply PCA to the 6-dimensional sensor readings across all time steps to visualize ingredient-level separability. The projection reveals visible clustering according to ingredient categories, particularly for spices and fruits, which occupy distinct regions in the PCA space. This suggests that sensor responses capture category-specific variance, potentially driven by differences in volatile compound profiles. However, we see that nuts and vegetables still largely overlap, which makes them hard to distinguish. Future work on better sensor designs and algorithmic advances can potentially provide stronger signals that improve models' performance in these two categories. To further investigate within-category separability, we performed a separate PCA only on fruits. In Fig. 2(c), individual fruits form well-separated clusters, e.g., peach, mandarin orange, lemon, and banana each exhibit distinct spatial groupings at a much better separation than the global categories. As a result, we expect the model to achieve better classification results on fruits categories as compared with nuts and vegetables.

Per-substance reading statistics are reported in Tab. 18, and kernel density estimation (KDE) plots for each category are shown in Figs. 16–21. Each sensor exhibits a distinct mean and standard deviation, indicating that preprocessing (Sec. 4.2) is necessary for stable prediction. The KDE plots also reveal category-specific response patterns across sensors. For example, the $C_2H_5OH$ channel tends to produce substantially higher readings for spices, which helps distinguish spices from other substance categories.

Fig. 3 describes the data distribution of SMELLNET-MIXTURE. In particular, with a binary mixture percentage of 77.9% and a ternary mixture percentage of 10%, the test subset provides a challenging environment for the model to predict the mixture ratios. As shown in Fig. 3(c), the mixtures span evenly across the entire substance space, with mixtures both within and across categories.

## 4 DEVELOPING SCENTFORMER FOR SMELL

Developing AI for smell poses challenges: sensor data is temporal, limited in quantity, and noisy. To address these, we design SCENTFORMER with a Transformer backbone, data-efficient training, and preprocessing to handle noise.

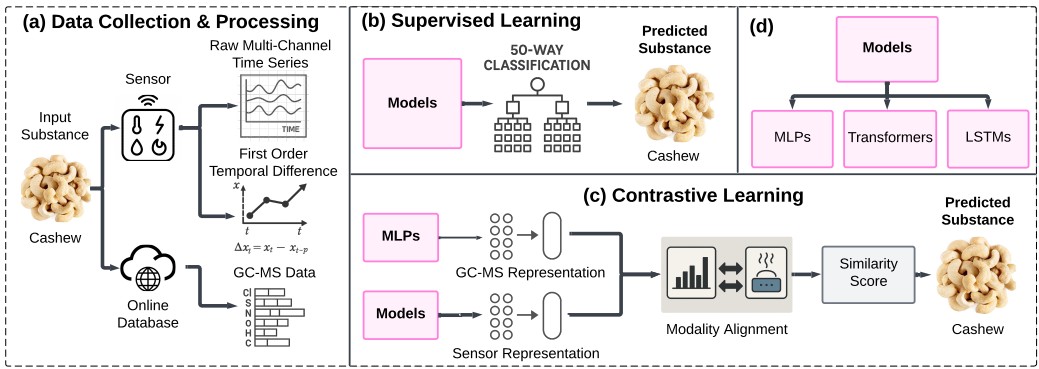

Figure 4: **Overview of the models used in this study.** (a) Raw multi-channel time series data is collected using a portable smell sensor as it samples an input substance (e.g., cashew). The data is optionally transformed using first-order temporal differences to emphasize signal dynamics. In parallel, high-resolution GC-MS data can be retrieved from an online database to provide chemical supervision. (b) In supervised learning, sensor data (raw or preprocessed) is passed through classification models trained to predict the correct substance among 50 classes. (c) In contrastive learning, paired sensor and GC-MS representations are aligned through modality-specific encoders. The resulting similarity scores rank the substance for prediction. (d) Our framework supports multiple model types—MLPs, LSTMs, and Transformers—each capable of ingesting either raw or temporally differenced sensor inputs to perform classification or representation learning.

## 4.1 PROBLEM SETUP AND NOTATION

Each example is a multichannel time series $x = (x_1, \ldots, x_T) \in \mathbb{R}^{T \times d}$. Our encoder $f_\theta$ produces an embedding $\mathbf{h} = f_\theta(x) \in \mathbb{R}^H$. For classification on SMELLNET-BASE, we map $x$ to a label $y \in \{1, \ldots, 50\}$. When using GC-MS data, we align $\mathbf{h}$ with a GC-MS embedding during training (App. E.1). For mixture distribution approximation on SMELLNET-MIXTURE, we map $x$ to a probability vector $\boldsymbol{\pi} \in [0, 1]^{12}$ with $\sum_{i=1}^{12} \pi_i = 1$. The target $\boldsymbol{\pi}^\star$ encodes the ground-truth mixture fractions (see App. E.2 for label construction).

## 4.2 SENSOR DATA PREPROCESSING

Given the temporal nature of the data, we apply the following preprocessing procedures:

**Temporal difference.** Unlike GC-MS data, which provide calibrated compound level concentrations, the MOX sensor channels provide uncalibrated real-valued responses that are best interpreted as *semi-quantitative* signals of the volatile profile. They are reliable for relative comparisons (e.g., stronger or weaker responses and temporal dynamics), but not as absolute concentration readouts without device specific calibration. To capture relative sensor changes, we apply a *temporal difference*. For each sensor channel $x_t$, we compute the difference over a fixed period of $p$ samples:

$$\Delta x_t = x_t - x_{t-p}, \quad \forall t > p.$$

When $p = 0$, we skip differencing and use the raw signal $x_t$.

**Sliding windows.** Due to the limited number of available recordings, we partition each file into smaller windows with size $w$ with a stride of $w/2$ to increase the effective dataset size. This strategy is popular in time-series domains to improve model generalization (Norwawi, 2021).

**Standardization.** All training data are aggregated to compute statistics for standardization, which are then applied to both training and evaluation sets. This ensures comparability across samples and reduces the influence of sensor noise.

## 4.3 SCENTFORMER ARCHITECTURE

We employ a pre-norm Transformer encoder over windowed sensor sequences, projected to a latent dimension $D$ and augmented with optional positional encodings and a learnable `[CLS]` token. Sequences are processed by stacked Transformer layers, with variable lengths supported via key–padding masks. The encoder output is pooled (mean or `[CLS]`) into a fixed-size vector. We attach a classification head for 50-way odor recognition, and two auxiliary heads for mixture presence and proportion prediction (see App. F.1 for implementation details).

## 4.4 TRAINING OBJECTIVES

We study three objectives: supervised classification from gas sensors, cross-modal alignment with GC-MS, and mixture ratio prediction.

**Supervised classification.** We train SCENTFORMER directly on SMELLNET-BASE windows for 50-way classification, using a softmax output and minimizing cross-entropy loss.

**Cross-modal alignment.** To leverage GC-MS data (FooDB Contributors, 2024; Liang et al., 2024), we adopt a symmetric contrastive learning objective (Radford et al., 2021; Socher et al., 2013) that aligns sensor and GC-MS embeddings (see App. F.2 for the full formula). We use GC-MS data as an ingredient-level chemistry prior rather than a per-sample sensor input. Concretely, for each ingredient we precompute a fixed GC-MS embedding from FooDB derived volatile-compound information. In the main paper we use a binned mass-spectral embedding ($X_{\mathrm{spec}}$), while App. C additionally compares a coarse elemental-composition embedding ($X_{\mathrm{atom}}$) as an alternative encoding. During cross-modal training, sensor-window embeddings are aligned to these ingredient-level GC-MS embeddings via a contrastive objective. At inference time, no new GC-MS measurement is required. Full details on GC-MS source coverage, embedding construction, and encoding ablations are provided in App. C.

**Mixture prediction.** For SMELLNET-MIXTURE windows, SCENTFORMER predicts normalized 12-D ratios. We optimize a composite loss combining KL divergence, hinge-$\ell_1$ penalty, and focal BCE (see App. F.3 for details).

## 5 EXPERIMENTS

We evaluate SCENTFORMER on SMELLNET for both single-substance classification and mixture distribution prediction. Specifically, we seek to answer the following research questions:

**RQ1:** To what extent can we classify single substance odors from SMELLNET-BASE alone, and which preprocessing choices contribute most to accuracy?
**RQ2:** Does cross-modal training with paired GC-MS data improve downstream sensor-only classification, and by how much?
**RQ3:** Can we predict the composition of mixtures from sensor time-series, and which preprocessing methods work best?

### 5.1 EVALUATION METRICS

For SMELLNET-BASE, we report Top-1 accuracy, Top-5 accuracy, macro-F1, and per-category accuracy. Top-1 and Top-5 measure whether the ground-truth ingredient appears in the top 1 or top 5 predicted classes, respectively. We report macro-F1 to provide a class-balanced summary that is less sensitive to class-frequency differences than accuracy alone.

For SMELLNET-MIXTURE, we evaluate predicted mixture proportions using mean absolute error (MAE), Top-1@0.1 accuracy, and a dynamic Top-$K$ hit rate. MAE measures the average absolute deviation between the predicted and ground-truth mixture proportions. For Top-1@0.1, a prediction is counted as correct if the predicted proportion is within ±0.1 of the ground-truth value on non-zero target components. We use the 0.1 threshold because mixture ratios in our dataset are defined on a 0.1 grid. Thus, ±0.1 corresponds to one step of the recipe resolution. Smaller thresholds would be below the precision of the mixing protocol, while substantially larger thresholds would be insufficiently discriminative for nearby ratios (e.g., 30:70 vs. 50:50). We additionally report a dynamic Top-$K$ hit rate, where $K$ equals the number of non-zero components in the target mixture (see App. I.2 for details). As supplementary mixture metrics, we also report KL divergence and cosine similarity between the predicted and ground-truth 12-D mixture distributions in the App. H.2.

### 5.2 EXPERIMENTAL SETUP

We evaluate SCENTFORMER on base-substance recognition and distribution prediction.

**SMELLNET-BASE:** For each ingredient, we use the last acquisition day (Day 6) as the held-out test day and train on the preceding five days. We compare SCENTFORMER against a non-temporal baseline (MLP) and temporal baselines (CNN and LSTM). To assess robustness to day-to-day acquisition shifts beyond this fixed last-day split, we additionally report a leave-one-day-out evaluation on SMELLNET-BASE in App. H.4.

Table 2: **Single-ingredient odor classification on SMELLNET-BASE (RQ1) and improvements from GC-MS integration (RQ2).** We vary preprocessing choices by window size ($w \in \{50, 100\}$) and period differencing ($p \in \{0, 25\}$). Temporal differencing ($p = 25$) yields large gains over raw signals, longer windows ($w = 100$) improve stability, and temporal models consistently outperform non-temporal baselines. Adding GC-MS supervision via contrastive learning further boosts weaker models but has mixed effects on temporal architectures. $\Delta$ Acc@1 reports the change in accuracy relative to the sensor-only baseline.

| Model | Window | p | Sensor-only (RQ1) | | | Cross-modal (RQ2) | | | $\Delta$ Acc@1 (X–S) |
|---|---|---|---|---|---|---|---|---|---|
| | | | Acc@1↑ | Acc@5↑ | F1↑ | Acc@1↑ | Acc@5↑ | F1↑ | |
| MLP | 50 | 0 | 21.9 | 55.8 | 17.2 | 24.1 | 58.3 | 20.7 | +2.2 |
| MLP | 50 | 25 | 18.2 | 49.0 | 17.8 | 24.7 | 58.7 | 24.5 | +6.5 |
| MLP | 100 | 0 | 21.0 | 54.4 | 17.4 | 24.3 | 57.2 | 20.4 | +3.3 |
| MLP | 100 | 25 | 26.8 | 59.7 | 24.7 | 30.8 | 63.7 | 28.6 | +4.0 |
| CNN | 50 | 0 | 25.5 | 61.2 | 23.3 | 27.7 | 69.9 | 23.8 | +2.2 |
| CNN | 50 | 25 | 46.9 | 81.7 | 46.1 | 45.7 | 82.5 | 44.8 | −1.2 |
| CNN | 100 | 0 | 29.5 | 66.6 | 24.9 | 26.7 | 71.3 | 24.0 | −2.8 |
| CNN | 100 | 25 | 52.7 | 85.6 | 50.5 | 58.9 | **88.4** | 57.0 | +6.2 |
| LSTM | 50 | 0 | 29.3 | 72.2 | 25.9 | 31.9 | 62.8 | 28.8 | +2.6 |
| LSTM | 50 | 25 | 50.6 | 84.7 | 48.8 | 50.6 | 79.5 | 49.6 | +0.0 |
| LSTM | 100 | 0 | 28.8 | 58.0 | 27.2 | 35.2 | 64.9 | 32.7 | +6.4 |
| LSTM | 100 | 25 | **57.9** | 87.0 | **56.0** | 56.9 | 85.2 | 55.2 | −1.0 |
| SCENTFORMER *(ours)* | | | | | | | | | |
| Transf. | 50 | 0 | 35.1 | 70.9 | 33.1 | 34.7 | 69.0 | 31.4 | −0.4 |
| Transf. | 50 | 25 | 50.6 | 85.0 | 49.5 | 52.6 | 82.5 | 51.7 | +2.0 |
| Transf. | 100 | 0 | 39.9 | 74.7 | 35.7 | 39.5 | 70.4 | 35.2 | −0.4 |
| Transf. | 100 | 25 | 56.1 | **87.4** | 55.5 | **63.3** | 86.1 | **61.7** | +7.2 |

To study temporal dynamics, we evaluate temporal differencing ($p \in \{0, 25\}$) and segment sensor streams into windows of size $w \in \{50, 100\}$. All models are trained for 90 epochs with batch size 32, using learning rates $\{3 \times 10^{-4}, 10^{-3}, 3 \times 10^{-3}\}$, and we report the final checkpoint for each configuration. We further investigate cross-modal training variants that incorporate paired GC-MS supervision via contrastive learning.

**SMELLNET-MIXTURE:** Mixture experiments use 12 odorants with both *seen* (novel sessions, known ratios) and *unseen* (zero-shot transfer) test splits. Models are trained with window sizes $w \in \{50, 100\}$, batch size 64, 60 epochs, and the same learning rate grid as SMELLNET-BASE.

Each SMELLNET-MIXTURE sample is defined by a volumetric recipe over 12 base odorants (e.g., binary and ternary mixtures), which we normalize to a 12-D target vector. The model predicts this continuous mixture distribution from the sensor time series. The goal is to recover the intended recipe proportions rather than exact gas-phase concentrations, which also depend on volatility and headspace dynamics.

Fig. 1 shows the hardware setup. Fig. 4 illustrates the evaluation pipeline, and full implementation details, additional hyperparameters, and reasons behind each choice are provided in App. G.2.

Table 3: **Distribution prediction on seen combinations (RQ3).** SCENTFORMER achieves the best overall performance, with the highest Top-1 and competitive Top-$K$ accuracy, highlighting the importance of temporal modeling for resolving overlapping odor signals.

| Model | Window | MAE ↓ | Top-1@0.1↑ | Top-$K$ (%)↑ |
|---|---|---|---|---|
| MLP | 50 | 0.0428 | 44.0 | 85.0 |
| | 100 | 0.0586 | 33.7 | 78.9 |
| CNN | 50 | 0.0404 | 48.1 | 86.7 |
| | 100 | 0.0476 | 36.2 | 87.0 |
| LSTM | 50 | 0.0399 | 46.4 | **89.3** |
| | 100 | 0.0430 | 46.5 | 86.3 |
| SCENTFORMER | 50 | **0.0395** | **50.2** | 87.9 |
| | 100 | 0.0417 | 47.9 | 89.0 |

### 5.3 RQ1: PREPROCESSING CHOICES EVALUATION

Based on the results in Tab. 2, we highlight three key findings:

**Finding 1.1: Temporal differencing substantially improves accuracy.** Adding temporal differencing ($p = 25$) mostly outperforms raw signals ($p = 0$), with an average gain of 16.1% across models and

window sizes. This demonstrates that temporal changes in sensor values carry critical discriminative information.

We do not find the gains from explicit temporal differencing surprising. Although temporal models can in principle learn lag-like transforms from raw sequences, time-series pipelines often use simple domain-motivated transforms to expose informative dynamics before learning (e.g., delta features in speech (Furui, 1986; Young et al., 2006), differencing and log-returns in finance (Tsay, 2010; Campbell et al., 1997)). For MOX sensors, absolute responses are affected by drift, device-specific offsets, and slow baseline shifts, whereas temporal changes more directly capture odor response dynamics. Consistent with this, temporal models already outperform MLP on raw signals ($p = 0$), and differencing further improves performance.

**Finding 1.2: Larger windows provide more stable patterns.** Window size $w = 100$ generally yields higher accuracy than $w = 50$, as longer temporal context captures more stable dynamics, though at the cost of fewer training and test samples. See App. I.1 for window calculation.

**Finding 1.3: Temporal models outperform non-temporal baselines.** Across preprocessing settings, CNN, LSTM, and SCENTFORMER consistently achieve higher accuracy than MLPs. This indicates that temporal structure in the sensor response carries discriminative information beyond static channel magnitudes. The result is consistent with Findings 1.1-1.2, which show that preprocessing choices that better expose temporal changes further improve performance.

### 5.4 RQ2: GC-MS INTEGRATION

Tab. 2 shows the effect of adding GC-MS supervision via contrastive learning.

**Finding 2.1: GC-MS supervision yields modest gains for weaker models.** Raw-signal inputs ($p = 0$) and non-temporal architectures see more of the consistent gains, showing that GC-MS embeddings provide complementary structure that compensates for limited model capacity.

**Finding 2.2: GC-MS supervision has mixed effects for stronger temporal models.** With GC-MS supervision, we see improvements can be large or slightly negative. Temporal architectures already capture much of the discriminative signal, but GC-MS further refines embeddings by grounding them in molecular structure, suggesting the two signals complement rather than replace each other.

**Finding 2.3: Effects depend on architecture and preprocessing.** In some cases (e.g., CNN at $w = 50, p = 25$), alignment brings little or even negative improvement, indicating that GC-MS can conflict with strong short-range features. This underscores that its value depends on how well the base model and preprocessing prepare features for cross-modal alignment.

Table 4: **Distribution prediction on unseen combinations with $w = 50$ (RQ3).** SCENTFORMER achieves the best overall performance, but accuracy drops substantially compared to the seen setting, highlighting the challenge of generalizing to novel odor mixtures.

| Model | Top-1@0.1↑ | Top-$K$ (%)↑ |
|---|---|---|
| MLP | 11.7 | 34.0 |
| CNN | 12.4 | 36.4 |
| LSTM | 11.8 | 34.2 |
| SCENTFORMER | 16.0 | 38.9 |

### 5.5 RQ3: MIXTURE PREDICTION

**Finding 3.1: SCENTFORMER outperforms other architectures on mixture prediction.** As shown in Tab. 3, SCENTFORMER achieves the best results across both Top-1@0.1 and Top-$K$ accuracy. This indicates that strong temporal modeling, which benefits single-substance recognition, is equally important for resolving overlapping signals in mixtures.

**Finding 3.2: Accuracy drops sharply for unseen mixtures.** Tab. 4 shows that performance degrades when evaluating on mixtures not seen during training. This suggests limited generalization: the model transfers poorly to novel ratios, even when all individual components were seen. Accuracy degradation indicates sensitivity to composition shift rather than merely class imbalance or session effects.

**Finding 3.3: Top-$K$ performance remains well above chance in unseen setting.** Although unseen mixtures are harder, SCENTFORMER still achieves substantially higher Top-$K$ accuracy than random guessing (around 16.7%, see App. I.2). This suggests that the learned representations encode meaningful compositional structure, enabling the model to narrow predictions to a plausible subset of substances even without explicit training on those mixtures.

These results show that SCENTFORMER is effective at mixture prediction, but generalization to unseen mixtures remains challenging. While temporal modeling provides clear benefits, scaling to

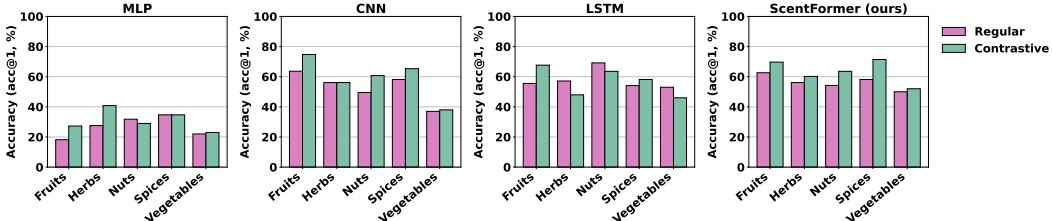

Figure 5: **Per-category accuracy** (acc@1) for four models at lag $p = 25$. Bars are paired per category (Regular vs. Contrastive). Non-temporal models underperform on vegetables, while temporal models are more robust across categories.

the combinatorial complexity of real-world odors may require compositional training strategies, data augmentation, or domain adaptation.

### 5.6 DISCUSSION & FUTURE WORK

To better understand the limitations of our models, we examine the per-category classification accuracy reported in Fig. 5. Several systematic patterns emerge. First, the non-temporal baseline shows pronounced weaknesses on certain categories, particularly vegetables. Vegetables exhibit substantial overlap with other categories in the PCA of sensor profiles (Fig. 2), making them harder to separate. MLP reaches relatively high accuracy on spices, whose sensor signatures are more distinct. This pattern is consistent with high within-class variance and overlapping volatile compound distributions.

In contrast, temporal architectures demonstrate stronger robustness across categories. By modeling temporal dynamics, these models capture transient variations in sensor readings that help differentiate harder classes. For example, SCENTFORMER maintains relatively balanced performance across all five categories, whereas MLP shows large category-specific disparities.

We also observe that contrastive training with GC-MS supervision provides differential benefits. For weaker models, alignment with molecular structure consistently improves recognition across categories, suggesting that the external chemical signal compensates for limited representational capacity. For temporal models, however, the effects are mixed, indicating that temporal modeling already captures much of the discriminative signal. This motivates future work on whether higher-resolution sensing or richer chemical supervision can provide additional gains.

A key limitation is that our data are collected in controlled settings without explicit background and environment subtraction across diverse ambient conditions, so performance may degrade under unseen environments, airflow patterns, or sensor placements. Future data collection should include explicit background subtraction protocols and broader cross-environment sampling to better evaluate out-of-domain robustness.

Additional ablations and channel-activity analyses are provided in App. H.

## 6 CONCLUSION

In this work, we introduced SMELLNET, a comparatively large benchmark for sensor-based machine olfaction using low-cost portable hardware. Built using portable and low-cost gas sensors, it captures over 828,000 data points across 50 base substances and 43 mixtures, paired with high-resolution GC-MS chemical data. SMELLNET establishes a benchmark for studying olfactory AI at scale, enabling both single-substance recognition and mixture prediction tasks. SMELLNET also inspires the design of SCENTFORMER, a Transformer-based architecture combining temporal differencing and sliding-window augmentation for smell data. We believe SMELLNET will serve as a foundation for future research in AI for smell and various real-world applications.

## 7 ETHICS STATEMENT

This work presents SMELLNET, a large-scale dataset for smell recognition using portable gas sensors. The dataset consists entirely of time-series sensor readings from chemical compounds in food substances and natural objects. No human subjects were involved in data collection, and the dataset contains no personally identifiable information. All data were collected using commercially available

sensors measuring volatile organic compounds from common food items purchased from public retailers.

Our collection system and classification models have minimal environmental impacts. Our sensor system is energy efficient, and can be powered by a USB cable with 5W input. The models are lightweight. On a single NVIDIA L40S (driver 550.54.14, CUDA 12.4) at batch size 32, SCENT-FORMER achieves mean per-window latency of 0.0191-0.0479 ms, as shown in App. H.6.

While SMELLNET is designed to advance research in olfactory AI with beneficial applications in food safety, healthcare, and environmental monitoring, we recognize that smell sensing technology could potentially be misused. We encourage responsible use of this dataset and the resulting models, particularly regarding privacy considerations in real-world deployments.

## 8    REPRODUCIBILITY STATEMENT

To support reproducibility, we provide comprehensive materials and documentation. The complete SMELLNET dataset (828,000 timesteps across 50 base substances and 43 mixtures) is available in the GitHub repository linked at the beginning of the paper, along with detailed instructions on data collection protocols and sensor specifications. Our sensor hardware configuration is fully documented in App. A, including circuit diagrams and component specifications. All preprocessing steps for the sensor time-series data are described in Sec. 4.2, with implementation details provided in our released codebase. Model hyperparameters are listed in Sec. 5.2 and App. F.1, App. G. For the GC-MS integration experiments, we provide the GC-MS preprocessing and descriptor-construction methods used in the paper, including the spectrum-based pipeline used in the main paper and an alternative pipeline in App. C. The repository also includes links to the public database used. The mixture label construction methodology is detailed in App. E.2, and the complete list of odorants and their sources is provided in Tab. 17. The dataset follows the hierarchical structure shown in Fig. 15, with CSV files organized by ingredient and recording session.

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

## A  SENSORS AND DATA COLLECTION ENVIRONMENT

### A.1  SENSOR HARDWARE SPECIFICATIONS

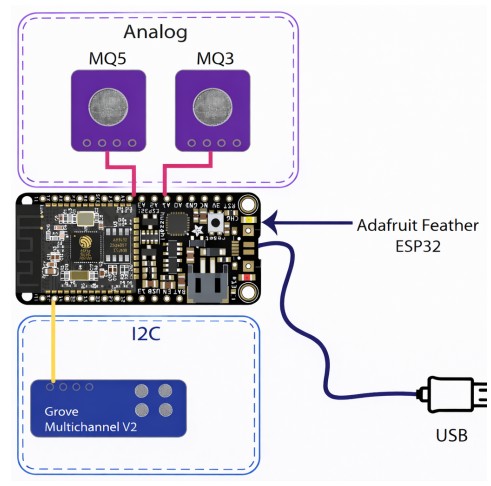

(a) Circuit diagram of sensor hardware setup.

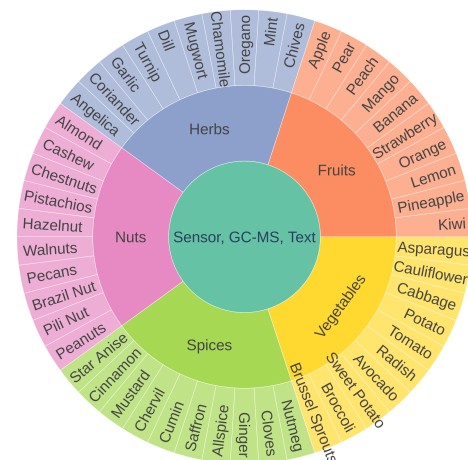

(b) Sunburst taxonomy of ingredient categories.

Figure 6: Overview of the SMELLNET dataset and sensing setup. (a) Our constructed portable smell sensor detects readings of various gases and atmospheric factors through 3 gas sensors. (b) SMELLNET includes smell sensor readings of 50 substances spanning nuts, spices, herbs, fruits, and vegetables.

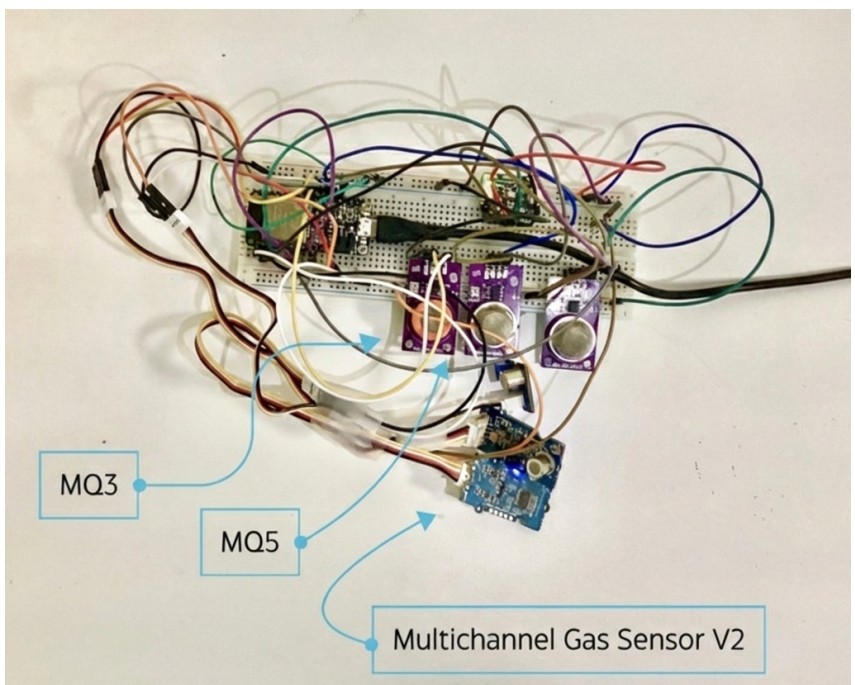

Figure 7: Sensor hardware architecture diagram of our prototype platform. The final camera-ready experiments use six gas-sensor channels from MQ-3, MQ-5, and the Grove Multichannel Gas Sensor V2 (manufacturer-labeled responses including CO, $NO_2$, $C_2H_5OH$, VOC, Alcohol, and LPG). Additional components shown in the schematic are legacy prototype sensors and are not used in the final reported experiments.

Our sensing device was constructed from a suite of commercially available gas sensors selected to provide broad coverage of volatile organic compounds (VOCs). Importantly, $C_2H_5OH$ and Alcohol refer to labels from two different physical sensor elements (with overlapping but non-identical cross-sensitivity), not two distinct target chemicals in our modeling. Fig. 7 shows the assembled sensor platform with all components labeled, and Tab. 5 lists the components used to build the device. We include these details to enable readers to reproduce the sensor used for data collection. Together,

Table 5: **Overview of smell sensors used in SMELLNET.** Datasheets: MQ-3 Hanwei Electronics Co., Ltd. (n.d.a), MQ-5 Hanwei Electronics Co., Ltd. (n.d.b), Grove Multichannel V2 Seeed Studio (n.d.).

| Sensor | Manufacturer | Description |
|---|---|---|
| MQ-3 | Hanwei Electronics | Metal-oxide (MOX) sensor optimized for detecting alcohol vapors. |
| MQ-5 | Hanwei Electronics | Detects LPG, natural gas, and coal gas (combustible gas detection). |
| Grove Multichannel V2 | Seeed Studio | Modular 4-channel MOX sensor measuring $NO_2$, CO, $C_2H_5OH$, and VOC. |

Table 6: Top feature contributions to the first two principal components (PC1 and PC2) of the sensor data. Features are sorted by contribution magnitude.

| Feature | PC1 | PC2 | Magnitude |
|---|---|---|---|
| LPG | 0.1514 | 0.7410 | 0.7563 |
| Alcohol | 0.1731 | 0.5548 | 0.5812 |
| $NO_2$ | 0.5044 | -0.2123 | 0.5473 |
| $C_2H_5OH$ | 0.5119 | -0.1683 | 0.5389 |
| VOC | 0.5035 | -0.1859 | 0.5367 |
| CO | 0.4206 | 0.1869 | 0.4603 |

these sensors provide complementary sensitivities and were integrated to maximize coverage of odor-related volatile signals in our dataset.

## A.2 CONTROLLED ENVIRONMENT

To ensure consistency and minimize external interference during data collection, all sensing sessions were conducted in a controlled environment. During each 10-minute recording interval, we placed both the food sample and the sensor array inside a transparent container. This enclosure prevented environmental factors such as airflow, human movement, or ambient contaminants from affecting the sensor readings. The container allowed gas emitted from the food to accumulate and diffuse evenly, while shielding the sensors from external disturbances such as changes in ambient composition caused by people walking nearby. Between sessions, we ventilated the enclosure to restore ambient conditions and eliminate residual smells from previous trials. After each session, we carefully monitor how the values change overall. Once all the sensor values are stable for 10 minutes, we consider the environment to have returned to ambient conditions, and we proceed to the next ingredient for the next session.

Despite these precautions, certain ambient factors, such as background $NO_2$ levels, varied across different days and could not be entirely eliminated.

## B SENSOR DATA

### B.1 MORE PCA

Fig. 8 shows PCA projections for nuts, spices, herbs, and vegetables. Across all categories, we observe distinct ingredient-level clustering, indicating that raw sensor signals inherently encode discriminative patterns. Notable examples include *potato* and *sweet potato* among vegetables, *dill* and *angelica* among herbs, and *nutmeg*, *star anise*, and *cumin* among spices. Even in the denser nuts category, ingredients like *almond* and *cashew* exhibit identifiable signatures.

These results support our hypothesis that portable gas sensors capture chemically meaningful variations, both across and within ingredient types, enabling fine-grained classification and motivating representation learning approaches.

Tab. 6 lists the top feature loadings for PC1 and PC2 of the 6-channel sensor readings. PC1 is driven primarily by $NO_2$, $C_2H_5OH$, VOC, and CO, while PC2 is dominated by LPG and Alcohol. This separation indicates that PCA captures distinct modes of variation across the gas-sensor channels, providing an interpretable summary of dominant sensor-response patterns for downstream analysis.

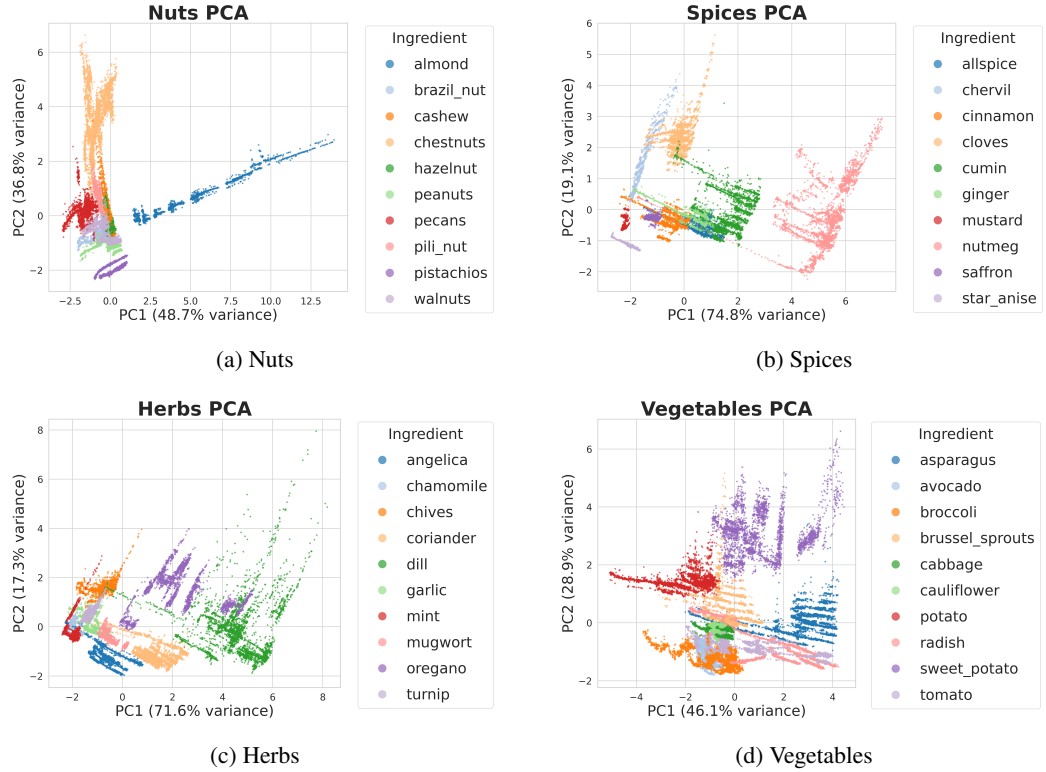

Figure 8: PCA projections of ingredient-level sensor responses for each major category. Each point represents a time step of raw sensor readings, colored by ingredient. Clear clusters are observed, indicating that sensor signals encode discriminative chemical signatures within each food category.

### B.2 FEATURE CORRELATION

Fig. 9 shows the Pearson correlation matrix for the 6 sensor channels. $NO_2$, $C_2H_5OH$, and VOC exhibit strong positive correlations ($r \approx 0.95$–$0.97$), indicating closely co-varying response patterns. CO is moderately correlated with this group ($r \approx 0.63$ to $0.67$), while Alcohol and LPG show weaker correlations with the $NO_2/C_2H_5OH$/VOC cluster and more independent behavior overall. These structured dependencies suggest partial redundancy among channels while still preserving complementary information for downstream modeling.

## C GC-MS DATA SOURCE AND REPRESENTATION

This appendix provides additional details on the GC-MS metadata used for cross-modal supervision, including data source and coverage, construction of ingredient-level GC-MS embeddings, and how GC-MS enters training and inference.

### C.1 GC-MS SOURCE AND COVERAGE

We do not collect GC-MS measurements in our sensing setup. Instead, we use FooDB derived volatile compound metadata as an ingredient-level chemistry prior. In our current experiments, all 50 base substances used in SMELLNET-BASE were selected to ensure FooDB coverage of volatile-compound annotations and associated GC-MS descriptors, yielding 100% coverage for the base-substance label set.

As a result, an ablation that removes ingredients without GC-MS descriptors is not meaningful for the current SMELLNET-BASE subset, because the class set would remain unchanged. Accordingly, the relevant comparison in our experiments is a controlled ablation on the same ingredients and evaluation protocol: sensor-only training (no GC-MS supervision) versus cross-modal training with GC-MS supervision.

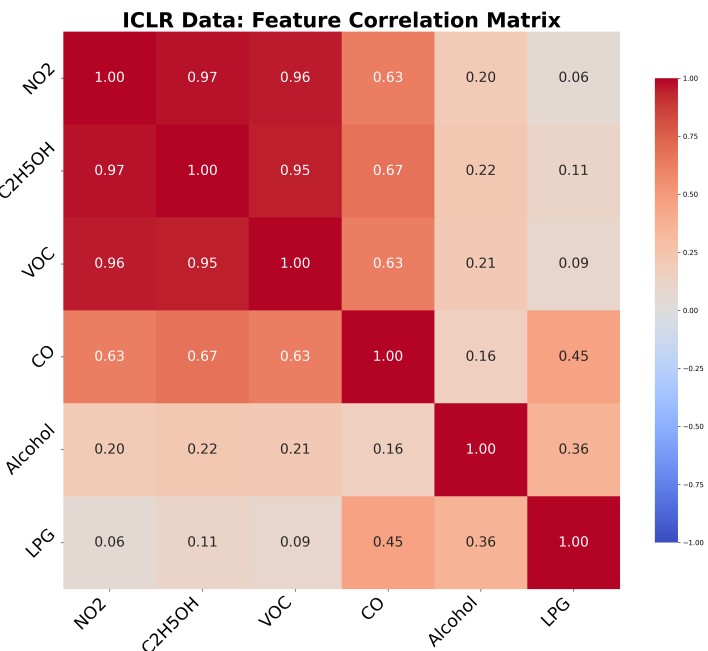

Figure 9: Pearson correlation matrix for the 6 sensor channels. $NO_2$, $C_2H_5OH$, and VOC are strongly correlated, while CO shows moderate correlation with this cluster. Alcohol and LPG are comparatively less correlated, indicating complementary sensor-response patterns.

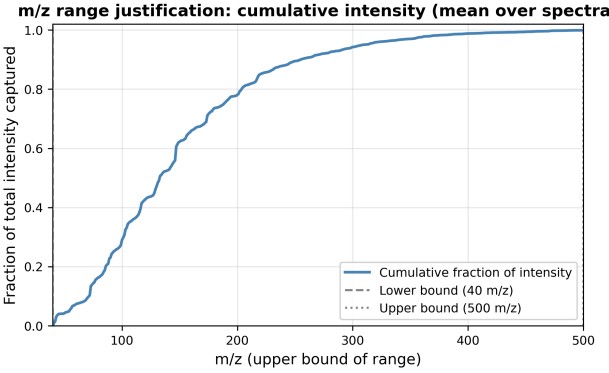

Figure 10: **Justification of the m/z range used for the binned mass-spectral embedding.** We plot the cumulative fraction of total spectral intensity captured as the upper m/z bound increases (mean over retrieved EI spectra). The curve rises rapidly at low-to-mid m/z values and saturates near 1.0 by 500 m/z, indicating that the selected 40–500 m/z range preserves nearly all intensity mass while keeping the embedding dimension compact. Dashed/dotted vertical lines mark the lower and upper bounds used in $X_{\text{spec}}$.

## C.2 GC-MS EMBEDDING CONSTRUCTION

For each ingredient, we precompute a fixed GC-MS embedding from its FooDB linked volatile compounds. We consider two ingredient-level GC-MS encodings.

**Binned mass-spectral embedding ($X_{\text{spec}}$: main-paper default).** For each ingredient, we retrieve its annotated volatile compounds from FooDB and collect available experimental Electron Ionization (EI) mass spectra for those compounds. Each spectrum is converted into a fixed-length vector of binned intensities over 40–500 m/z using 1 Da bins, and each spectrum is max-normalized. We then average spectra for the same compound and subsequently average across compounds to obtain a single fixed-length GC-MS embedding for the ingredient. We use the 40–500 m/z range because it captures

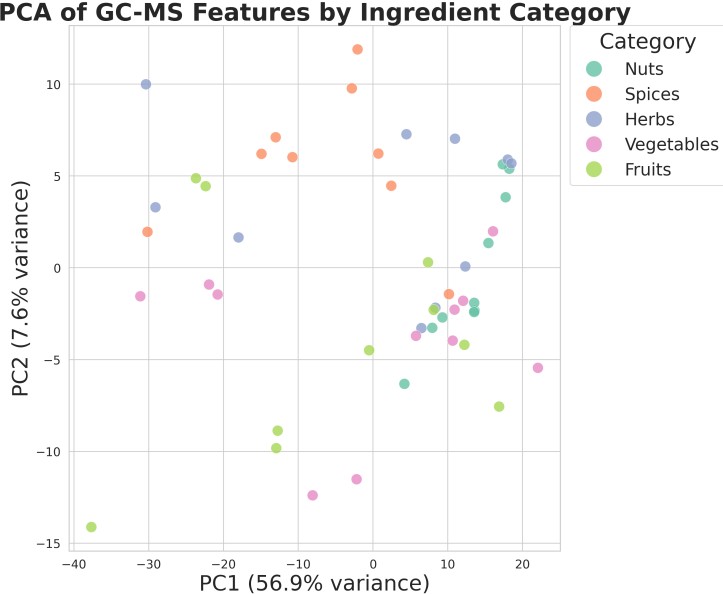

Figure 11: **PCA of ingredient-level GC-MS fingerprints** ($X_{\text{spec}}$)**, colored by category.** Each point corresponds to one ingredient embedding constructed from averaged, binned EI spectra. The first two principal components reveal both category-level organization and within-category spread, indicating that the binned mass-spectral representation captures meaningful chemical variation across foods. Axes report the explained variance of each principal component.

the vast majority of signal intensity in our retrieved EI spectra while keeping the representation compact as shown in Fig. 10.

**Elemental-composition embedding ($X_{\text{atom}}$: alternative encoding).** As an alternative descriptor, we also construct an ingredient-level embedding from elemental-composition statistics derived from the molecular formulas of the ingredient's annotated volatile compounds. This representation captures coarse compositional trends but discards fine-grained spectral structure. We include it as an alternative encoding for robustness analysis and comparison with earlier versions of the paper.

### C.3   How GC-MS is used in cross-modal training

GC-MS information enters the model only through *ingredient-level embeddings*. During training, we apply the symmetric contrastive objective described in App. F.2 to align sensor-window embeddings with the GC-MS embedding of the corresponding ingredient while pushing away mismatched ingredient embeddings. Thus, GC-MS acts as a chemistry-aware label representation (or prior) during training.

At inference time, no new GC-MS measurement is required. We embed the sensor window using the trained sensor encoder and perform classification using the learned label-aligned representation. Therefore, GC-MS is not used as an additional per-sample input modality at test time.

### C.4   Qualitative structure of GC-MS embeddings

We next examine the qualitative structure of the ingredient-level GC-MS embeddings. For the main-paper representation $X_{\text{spec}}$, we perform PCA on the binned mass-spectral embeddings. Figure 11 shows the projection onto the first two principal components, with points colored by ingredient category. The projection exhibits both category-level organization and substantial within-category variation, suggesting that $X_{\text{spec}}$ captures meaningful chemical differences across ingredients while preserving fine-grained structure beyond coarse compositional summaries.

Fig. 12 shows examples of ingredient-level GC-MS fingerprints (i.e., the averaged binned EI spectra used to construct $X_{\text{spec}}$). These examples illustrate that $X_{\text{spec}}$ retains interpretable peak patterns across m/z bins while providing a fixed-length representation suitable for contrastive alignment with sensor embeddings.

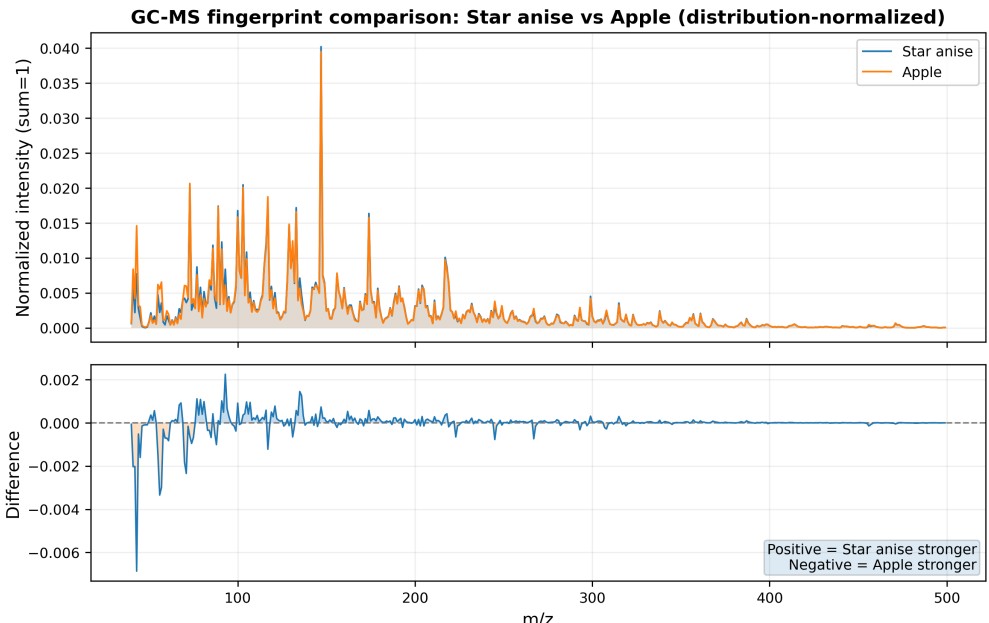

Figure 12: **Example comparison of ingredient-level GC-MS fingerprints in the binned spectral space.** Top: distribution-normalized GC-MS fingerprints for *star anise* and *apple*, shown as averaged binned EI mass spectra over the selected m/z range. Bottom: pointwise difference between the two normalized fingerprints (star anise minus apple). Although the overall spectral envelopes are similar, the difference plot reveals localized peak-intensity deviations across multiple m/z bins, illustrating that visually similar fingerprints can still encode discriminative ingredient-specific structure in $X_{\text{spec}}$.

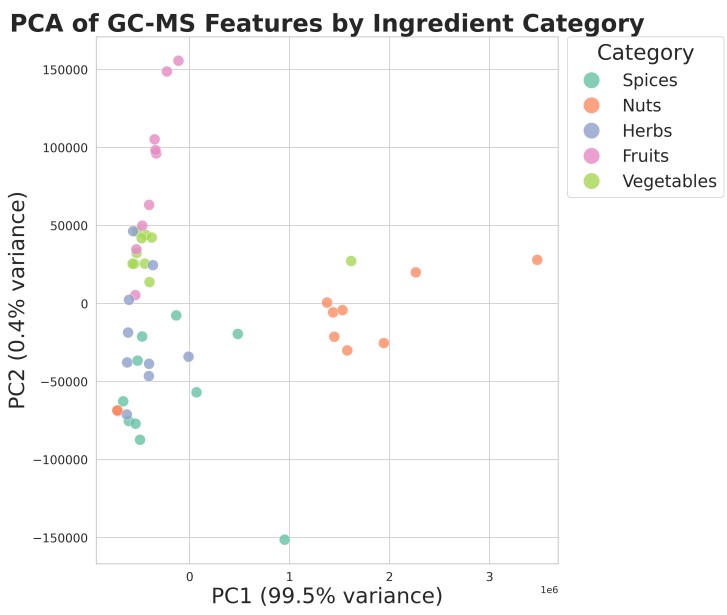

Figure 13: PCA of GC-MS elemental composition across ingredient categories. PC1 accounts for 99.5% of the variance, highlighting dominant compositional differences (e.g., carbon and hydrogen levels).

For completeness, we also examine the alternative elemental-composition embedding $X_{\text{atom}}$. Although $X_{\text{atom}}$ is substantially coarser than $X_{\text{spec}}$, it still exhibits non-trivial structure across ingredients, reflecting broad compositional trends in the underlying volatile compounds. Fig. 13 shows the PCA projection of $X_{\text{atom}}$, which reveals category-level variation driven by dominant elemental

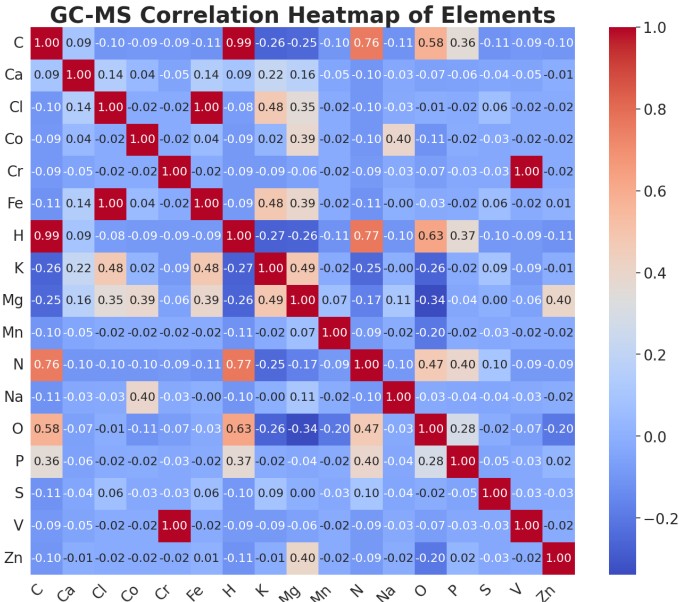

Figure 14: GC-MS correlation heatmap of elemental counts. Strong positive correlations are observed between common organic elements (C, H, N, O), while many trace elements are uncorrelated.

statistics. Fig. 14 further shows the element-wise correlation heatmap, highlighting interpretable dependencies among common elements in organic molecules. These diagnostics help explain why $X_{atom}$ can still provide useful cross-modal supervision despite discarding fine-grained spectral peak information. At the same time, its reduced expressiveness motivates our use of $X_{spec}$ as the default GC-MS representation in the main paper.

## D  SMELLNET OVERVIEW

Table 7: Descriptive statistics of sensor readings in the training dataset (150,711 samples).

|  | NO$_2$ | C$_2$H$_5$OH | VOC | CO | Alcohol | LPG |
|---|---|---|---|---|---|---|
| Mean | 97.80 | 138.33 | 195.94 | 792.94 | 3.41 | 30.33 |
| Std | 118.68 | 143.40 | 196.19 | 61.67 | 3.28 | 38.29 |
| Min | 13.00 | 39.00 | 26.00 | 705.00 | 0.00 | 2.00 |
| 25% | 35.00 | 65.00 | 73.00 | 750.00 | 1.00 | 14.00 |
| 50% | 46.00 | 77.00 | 106.00 | 776.00 | 2.00 | 23.00 |
| 75% | 105.00 | 140.00 | 232.50 | 820.00 | 5.00 | 32.00 |
| Max | 753.00 | 863.00 | 953.00 | 1006.00 | 42.00 | 507.00 |

Table 8: Descriptive statistics of sensor readings in the testing dataset (29,423 samples).

|  | NO$_2$ | C$_2$H$_5$OH | VOC | CO | Alcohol | LPG |
|---|---|---|---|---|---|---|
| Mean | 92.33 | 134.08 | 187.80 | 791.69 | 3.47 | 33.31 |
| Std | 113.90 | 140.97 | 191.23 | 60.11 | 3.14 | 46.17 |
| Min | 15.00 | 41.00 | 26.00 | 710.00 | 0.00 | 3.00 |
| 25% | 34.00 | 65.00 | 72.00 | 750.00 | 1.00 | 15.00 |
| 50% | 45.00 | 73.00 | 93.00 | 774.00 | 2.00 | 23.00 |
| 75% | 95.00 | 135.00 | 218.00 | 823.00 | 6.00 | 35.00 |
| Max | 775.00 | 850.00 | 947.00 | 1004.00 | 30.00 | 444.00 |

### D.1  SMELLNET SUMMARY

This appendix provides descriptive statistics for all 6 sensor channels in both the training and testing datasets. Tab. 7 and Tab. 8 summarize key distributional properties, including mean, standard deviation, and range for each feature. The sensor readings exhibit substantial variability across

samples, particularly in channels such as VOC and $NO_2$. These statistics highlight the diversity and dynamic range of the collected data, which underpin the challenges of robust model generalization in real-world settings. We include a text description generated by LLMs of all substances we used for future experiments to enable alignment between text and smell modalities.

## D.2 DATASET HIERARCHY

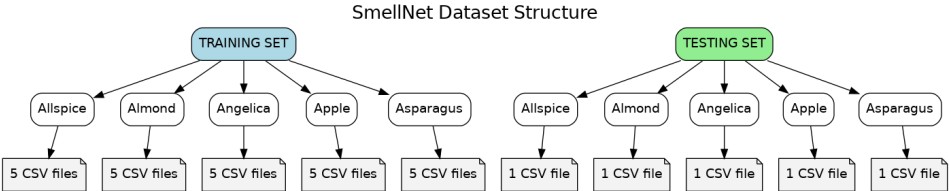

Figure 15: Hierarchical organization of the SMELLNET-BASE dataset. Each ingredient folder contains multiple CSV files with raw sensor time series data.

Fig. 15 illustrates the hierarchical structure of the SMELLNET-BASE dataset. Each ingredient is represented as a folder containing multiple time series recordings in CSV format. The training set includes five CSV samples per ingredient to capture variation across trials, while the testing set contains one representative CSV file per ingredient. This structure ensures a consistent, per-ingredient organization and facilitates reproducible supervised learning.

## E ADDITIONAL DETAILS FOR PROBLEM SETUP AND NOTATION

### E.1 GC-MS DESCRIPTOR CONSTRUCTION

When a GC-MS readout is profiled for a substance class, we construct a fixed-length descriptor $g \in \mathbb{R}^{d'}$ by aggregating element counts over a fixed set $\mathcal{E}$ (e.g., C, H, O, N, S, Cl). We then apply per-dimension standardization using training-set statistics:

$$\tilde{g}_j = \frac{g_j - \mu_j}{\sigma_j}, \qquad j = 1, \dots, d',$$

and use $\tilde{g}$ in all GC-MS–aware objectives.

### E.2 MIXTURE-LABEL CONSTRUCTION AND EVALUATION METRICS

**Targets.** Each example in SMELLNET-MIXTURE has a ground-truth composition over $K = 12$ odorants. Let $\tilde{\pi} \in \mathbb{R}^{K}_{\geq 0}$ denote raw proportions, we normalize to the probability:

$$\pi^{\star} = \frac{\tilde{\pi}}{\sum_{i=1}^{K} \tilde{\pi}_i}.$$

The model predicts $\pi = g_\theta(x)$ via a softmax head.

## F BUILDING SCENTFORMER

### F.1 DETAILED SCENTFORMER ARCHITECTURE

**Backbone.** Input windows $x' \in \mathbb{R}^{w \times d}$ are linearly projected to dimension $D$. We use sinusoidal positional encodings and prepend a learnable [CLS] token. The model comprises $L$ pre-norm Transformer layers with $H$ attention heads, feed-forward width $4D$, and dropout probability $p_{\text{drop}}$.

**Pooling and heads.** Given encoder output $H \in \mathbb{R}^{w' \times D}$, we apply masked mean pooling (default) or [CLS] embedding to obtain $h \in \mathbb{R}^D$. A two-layer MLP projects $h$ to 50 logits. Linear heads predict mixture presence $\hat{u} \in \mathbb{R}^{12}$ and proportions $\hat{z} = \text{softmax}(W_m h + b_m) \in \Delta^{11}$.

### F.2 CONTRASTIVE LEARNING LOSS

Given $N$ gas sensor embeddings $z_i^{(s)}$ and corresponding GC-MS embeddings $z_i^{(g)}$, we minimize:

$$\mathcal{L}_{\text{contrastive}} = -\frac{1}{N} \sum_{i=1}^{N} \left[ \log \frac{\exp(\text{sim}(z_i^{(s)}, z_i^{(g)})/\tau)}{\sum_j \exp(\text{sim}(z_i^{(s)}, z_j^{(g)})/\tau)} + \log \frac{\exp(\text{sim}(z_i^{(g)}, z_i^{(s)})/\tau)}{\sum_j \exp(\text{sim}(z_i^{(g)}, z_j^{(s)})/\tau)} \right],$$

where $\text{sim}$ is cosine similarity and $\tau$ a temperature.

### F.3 Mixture Prediction Objective

**Intuition.** Our loss for mixture prediction blends three complementary terms to balance proportion accuracy, robustness, and class imbalance. *(i) KL divergence* encourages the predicted distribution $\hat{p} = \mathrm{softmax}(z)$ to match the ground-truth proportions $p$. *(ii) A hinge-$\ell_1$ penalty* is applied only to components that are truly present, tightening errors beyond a small tolerance $\varepsilon$. *(iii) Focal binary cross-entropy (Focal BCE)* operates on presence/absence labels and down-weights easy negatives while focusing learning on hard positives. Scalars $\alpha$ and $\beta$ balance the second and third terms relative to KL.

**Full objective.** Let $r_i = \mathbf{1}[p_i > 0]$ indicate presence, and $S = \{i : r_i = 1\}$ the set of present components. The loss is

$$\mathcal{L} = \mathrm{KL}(p \,\|\, \hat{p}) + \alpha \frac{1}{|S|} \sum_{i \in S} \max\big(|\hat{p}_i - p_i| - \varepsilon, 0\big) + \beta \cdot \mathrm{FocalBCE}(s, r),$$

with $\hat{p} = \mathrm{softmax}(z)$, and FocalBCE using $(\alpha_f{=}0.75, \gamma{=}2.0)$ in our experiments. Here, $\mathrm{KL}(p\|\hat{p})$ promotes globally accurate proportions, the hinge-$\ell_1$ term tightens errors on present components beyond tolerance $\varepsilon$, and the focal term addresses class imbalance in presence prediction.

## G Additional Experimental Setup

### G.1 Model details

This section specifies input/shape conventions, pooling/masking semantics, and per-architecture hyperparameters used in our code. We purposefully omit high-level architecture and objective overviews that appear in the main text and earlier appendices.

**Input & shapes.** Unless noted otherwise, models consume windows $x \in \mathbb{R}^{B \times T \times F}$ (batch, time, features). For variable-length batches, we pass sequence lengths $\ell \in \mathbb{N}^B$. When a layer expects channel-first, we convert to $(B, C, T)$ internally.

**Mask semantics.** Where masks are used, *True* marks *padding*. For mean pooling over valid tokens we use

$$\bar{h}_b = \frac{\sum_t m_{b,t}\, h_{b,t}}{\max\big(\sum_t m_{b,t},\, 10^{-6}\big)}, \quad m_{b,t} = \mathbf{1}[t < \ell_b],$$

and apply the same $m$ to exclude pad tokens from attention or max-pool operations.

**ScentFormer**

- **Input stem:** Linear($F \to D$) then LayerNorm, optional sinusoidal positional encodings added in-place.
- **Tokenization:** Optional `[CLS]` (learnable, $\mathcal{N}(0, 0.02^2)$). Pooling is either masked mean (default) or `[CLS]`.
- **Encoder:** $L$ pre-norm layers with $H$ heads, FFN width $4D$, dropout $p$, activation `gelu`.
- **Head:** Linear($D \to D/2$)–GELU–Dropout–Linear($D/2 \to C$).
- **Defaults:** $H{=}8$, $L{=}4$, $p_{\mathrm{drop}}{=}0.1$, activation=`gelu`, positional enc.=on, `[CLS]`=off, pool=mean.

**LSTMNet**

- **Core:** LSTM($F \to H$) with $L$ layers, bidirectional by default. Dropout $p$ only if $L > 1$.
- **Pooling:** `last` (concat final fwd/bwd), masked `mean`, or masked `max`.
- **Variable length:** Uses `pack_padded_sequence` / `pad_packed_sequence`.
- **Projection & head:** Linear($\cdot \to D_{\mathrm{emb}}$) then Linear($D_{\mathrm{emb}} \to C$).
- **Defaults:** $L{=}1$, bidirectional, $p_{\mathrm{drop}}{=}0.1$, pool=mean.

**CNN1D classifier**

- **Layout:** Stack of

  `Conv1d`($C_{\mathrm{in}} \to C_{\mathrm{out}}$, same padding via $k//2$)–BatchNorm–ReLU–(Dropout).

- **Head:** Global average pooling over $T$ then Linear($C' \to C$).
- **Channel order:** Accepts $(B, T, C)$ (`channel_last=true`) or $(B, C, T)$. We coerce to $(B, C, T)$ internally.
- **Defaults:** channels=(64,128,256), kernel size $k$=5, $p_{\text{drop}}$ = 0.2, BatchNorm on, `channel_last=true`.

**MLP classifier**

- **Pooling:** If input is $(B, T, C)$ or $(B, C, T)$, pool over $T$ via mean/max (default: mean). Optionally `flatten` requires fixed $T$ with input dim $C \times T$.
- **Backbone:** Repeated [Linear–(BatchNorm)–ReLU–(Dropout)] blocks. Head is Linear$\to C$.
- **Defaults:** hidden sizes (256,256), BatchNorm on, $p_{\text{drop}}$ = 0.2, pool=mean, `channel_last=true`.

**GC-MS MLP encoder**

- **Stem:** Optional LayerNorm on input, then MLP with ReLU and optional BatchNorm/Dropout, ending in Linear$\to D$.
- **Normalization:** Optional $\ell_2$ normalization of the final embedding when used in contrastive objectives.
- **Defaults:** hidden (512,256), $D$=256, $p_{\text{drop}}$ = 0.1, LayerNorm on, BatchNorm off, $\ell_2$ off.

**Hyperparameters**

| Component | Key defaults / toggles |
|---|---|
| Transformer | $H$=8, $L$=4, FFN = $4D$, $p_{\text{drop}}$=0.1, GELU, PE on, CLS off, pool=mean |
| LSTMNet | $L$=1, bi=True, $p_{\text{drop}}$=0.1, pool $\in$ {mean,last,max}, $D_{\text{emb}}$ as set |
| CNN1D | channels=(64,128,256), $k$=5, BN on, $p_{\text{drop}}$ = 0.2, `channel_last=true` |
| MLP | hidden=(256,256), BN on, $p_{\text{drop}}$ = 0.2, pool $\in$ {mean,max,flatten} |
| GC-MS enc. | hidden=(512,256), $D$=256, LayerNorm on, BN off, dropout 0.1, $\ell_2$ off |

### G.2 TRAINING HYPERPARAMETERS

We standardize training across baselines and SCENTFORMER to ensure a fair comparison and to avoid overfitting to any one configuration.

**Temporal differencing.** To quantify the value of short-range dynamics, we evaluate fixed lags $p \in \{0, 25\}$ when forming first-order temporal differences $\Delta x_t = x_t - x_{t-p}$ (Sec. 4.2). This follows prior evidence in our setting that differencing can substantially improve discriminability.

**Sliding-window segmentation.** We segment streams into overlapping windows of length $w \in \{50, 100\}$ (Sec. 4.2). The shorter window ($w = 50$) increases the number of training and evaluation samples, whereas the longer window ($w = 100$) trades sample count for more stable temporal context (App. I.1).

**Learning rate selection.** To keep model comparisons robust and reproducible, we tune over a small, fixed grid shared by all methods: $\{3 \times 10^{-4}, 10^{-3}, 3 \times 10^{-3}\}$. We evaluate the final checkpoint of each run on the validation set and choose the best final checkpoint. Limiting the grid prevents "hyperparameter fishing" and reduces variance attributable to optimizer settings.

**Epoch budgets and batching.** For SMELLNET-BASE classification we train for 90 epochs with batch size 32. For SMELLNET-MIXTURE distribution prediction we train for 60 epochs with batch size 64. Using fixed epoch budgets across models minimizes variance due to training length. The best checkpoint is chosen by validation Top-1.

**Randomization.** We fix the Python-level random seed to 42 for reproducibility across all experiments.

**GC-MS supervision.** Contrastive alignment with GC-MS is used only for the single-ingredient classification setting, where per-ingredient GC-MS signals are available. We do not apply GC-MS supervision to mixture prediction because reliable GC-MS profiles for arbitrary mixtures are not available at scale.

Table 9: **GC-MS encoding ablation on SMELLNET-BASE (Acc@1, %).** We compare sensor-only training ($S$), cross-modal training with an elemental-composition GC-MS embedding ($X_{\mathrm{atom}}$), and cross-modal training with a binned mass-spectral GC-MS embedding ($X_{\mathrm{spec}}$). The sensor encoder, training protocol, and evaluation split are unchanged across columns for each row. $\Delta_{\mathrm{atom}}$ and $\Delta_{\mathrm{spec}}$ denote the Acc@1 change relative to $S$ under the same model and preprocessing setting.

| Model | $w$ | $p$ | Acc@1 (%) | | | $\Delta$ vs. $S$ | |
|---|---|---|---|---|---|---|---|
| | | | $S$ | $X_{\mathrm{atom}}$ | $X_{\mathrm{spec}}^{*}$ | $\Delta_{\mathrm{atom}}$ | $\Delta_{\mathrm{spec}}^{*}$ |
| MLP | 50 | 0 | 21.9 | 23.6 | 24.1 | +1.7 | +2.2 |
| | 50 | 25 | 18.2 | 23.8 | 24.7 | +5.6 | +6.5 |
| | 100 | 0 | 21.0 | 25.6 | 24.3 | +4.6 | +3.3 |
| | 100 | 25 | 26.8 | 28.0 | 30.8 | +1.2 | +4.0 |
| CNN | 50 | 0 | 25.5 | 28.4 | 27.7 | +2.9 | +2.2 |
| | 50 | 25 | 46.9 | 45.9 | 45.7 | −1.0 | −1.2 |
| | 100 | 0 | 29.5 | 31.0 | 26.7 | +1.5 | −2.8 |
| | 100 | 25 | 52.7 | 57.1 | 58.9 | +4.4 | +6.2 |
| LSTM | 50 | 0 | 29.3 | 29.7 | 31.9 | +0.4 | +2.6 |
| | 50 | 25 | 50.6 | 53.3 | 50.7 | +2.7 | +0.1 |
| | 100 | 0 | 28.8 | 33.7 | 35.2 | +4.9 | +6.4 |
| | 100 | 25 | 57.9 | 56.1 | 56.9 | −1.8 | −1.0 |
| Transformer | 50 | 0 | 35.1 | 36.2 | 34.7 | +1.1 | −0.4 |
| | 50 | 25 | 50.6 | 50.9 | 52.6 | +0.3 | +2.0 |
| | 100 | 0 | 39.9 | 41.0 | 39.5 | +1.1 | −0.4 |
| | 100 | 25 | 56.1 | 58.5 | 63.3 | +2.4 | +7.2 |

Note. $X_{\mathrm{spec}}^{*}$ denotes the binned mass-spectral GC-MS embedding.

**Design choice for mixtures (no temporal differencing).** For SMELLNET-MIXTURE, sensor streams are recorded at 10 Hz on a four-channel array. At this sampling rate, small lags yield negligible signal change, whereas larger lags substantially reduce the effective number of windows. We therefore train mixture models on raw (non-differenced) windows.

# H ADDITIONAL EXPERIMENT RESULTS AND ABLATION STUDY

## H.1 ENCODING ABLATION: $X_{\mathrm{atom}}$ VS. $X_{\mathrm{spec}}$

To test whether our RQ2 conclusions depend on the specific GC-MS encoding, we compare three settings on SMELLNET-BASE: sensor-only training (denoted $S$, no GC-MS supervision), cross-modal training with the elemental-composition GC-MS embedding ($X_{\mathrm{atom}}$), and cross-modal training with the binned mass-spectral GC-MS embedding ($X_{\mathrm{spec}}$). The sensor encoder, training schedule, and evaluation protocol are unchanged. Only the GC-MS-side ingredient embedding is replaced.

Tab. 9 reports Acc@1 for each architecture, window size $w$, and temporal differencing lag $p$. We also report $\Delta_{\mathrm{atom}}$ and $\Delta_{\mathrm{spec}}$, defined as the change in Acc@1 relative to the sensor-only baseline $S$ under the same architecture and preprocessing setting.

Overall, $X_{\mathrm{spec}}$ shows the same qualitative behavior as $X_{\mathrm{atom}}$: GC-MS supervision yields the largest gains for weaker models (especially MLP), while stronger temporal models (CNN, LSTM, and SCENTFORMER) exhibit smaller and sometimes mixed effects depending on preprocessing. This confirms that our main RQ2 conclusions are robust to the choice of GC-MS encoding. In both cases, GC-MS functions as a complementary chemistry-aware prior that modestly regularizes the sensor embedding, while the dominant performance gains come from modeling the sensor time series itself.

## H.2 ADDITIONAL MIXTURE METRICS AND FULL RESULTS

Each SMELLNET-MIXTURE sample is defined by a volumetric recipe over 12 base odorants, normalized to a 12-dimensional target vector. The model predicts this continuous mixture distribution directly from the sensor time series, with the goal of recovering the intended recipe proportions rather than exact gas-phase concentrations, which additionally depend on volatility and headspace dynamics.

Table 10: **Mixture proportion prediction performance on SMELLNET-MIXTURE.** We report KL divergence, MAE, and cosine similarity between the predicted and ground-truth 12-D mixture distributions for seen and unseen mixtures across models and window sizes.

| Split | $w$ | Model | KL↓ | MAE↓ | Cosine↑ |
|---|---|---|---|---|---|
| Seen | 50 | MLP | 0.450 | 0.0428 | 0.861 |
| | 50 | CNN | **0.370** | 0.0404 | 0.867 |
| | 50 | LSTM | 0.416 | 0.0399 | **0.878** |
| | 50 | SCENTFORMER | 0.480 | **0.0395** | 0.863 |
| | 100 | MLP | 0.615 | 0.0586 | 0.799 |
| | 100 | CNN | 0.425 | 0.0476 | 0.849 |
| | 100 | LSTM | 0.503 | 0.0430 | 0.856 |
| | 100 | SCENTFORMER | 0.386 | 0.0417 | 0.874 |
| Unseen | 50 | MLP | 2.919 | 0.1190 | 0.860 |
| | 50 | CNN | 2.649 | 0.1160 | 0.860 |
| | 50 | LSTM | 2.532 | 0.1160 | 0.848 |
| | 50 | SCENTFORMER | 2.618 | **0.1100** | 0.863 |
| | 100 | MLP | **2.302** | 0.1110 | 0.799 |
| | 100 | CNN | 2.488 | 0.1160 | 0.824 |
| | 100 | LSTM | 2.441 | 0.1150 | 0.856 |
| | 100 | SCENTFORMER | 2.454 | 0.1110 | **0.895** |

Table 11: Single-channel mask ablation (SCENTFORMER, window=50, temporal difference $p$=25). Entries are *negative* point changes in Acc@1 when masking a channel (masking always reduces accuracy). Baselines: Cross-modal: 52.60%, Sensor-only: 50.60%.

| Cross-modal | | Sensor-Only | |
|---|---|---|---|
| Channel (gas) | $\Delta$ Acc@1 | Channel (gas) | $\Delta$ Acc@1 |
| LPG | $-28.07$ | LPG | $-28.90$ |
| VOC | $-24.65$ | Alcohol | $-26.50$ |
| Alcohol | $-23.92$ | VOC | $-24.10$ |
| CO | $-18.93$ | CO | $-22.07$ |
| $C_2H_5OH$ | $-17.36$ | $NO_2$ | $-19.76$ |
| $NO_2$ | $-15.60$ | $C_2H_5OH$ | $-10.25$ |

In addition to Top-1@0.1 and dynamic Top-$K$ accuracy, we report KL divergence, MAE, and cosine similarity between the predicted and ground-truth 12-D mixture distributions. We use Top-1@0.1 because mixture ratios are defined on a 0.1 grid. A ±0.1 tolerance therefore corresponds to exactly one step of the recipe resolution.

Tab. 10 reports full mixture prediction results for seen and unseen mixtures across models and window sizes. Cosine similarity remains relatively high in both settings, while KL divergence and MAE increase substantially on unseen mixtures. This divergence between metrics indicates that models tend to recover the *direction* of the mixture vector, while still incurring substantial absolute errors in the predicted proportions for unseen combinations. We therefore treat cosine similarity as a useful complementary diagnostic, whereas MAE, KL divergence, and Top-1@0.1 remain the primary metrics for evaluating ratio accuracy.

### H.3 CHANNEL IMPORTANCE THROUGH MASKING

All reported values are negative $\Delta$Acc@1, i.e., masking any single channel reduces Top-1 accuracy. With temporal differencing ($p$=25, Tab. 11), the largest drops occur for LPG ($-28.9\%$ to $-28.1\%$), followed by VOC and Alcohol, indicating these channels carry the most decisive information under different preprocessing. Without differencing ($p$=0, Tab. 12), $NO_2$ and CO dominate the impact (up to $-25.2\%$), suggesting the raw-signal model leans more on oxidizing and CO-related responses.

Contrastive training slightly reshapes importance but preserves the main ordering within each pre-processing regime. Overall, the strictly negative $\Delta$Acc@1 across all cells reinforces that each gas channel contributes uniquely. The magnitude pattern shifts with temporal preprocessing and contrastive alignment.

Table 12: Single-channel mask ablation (SCENTFORMER, window=50, *no* temporal difference $p=0$). Entries are *negative* point changes in Acc@1 when masking a channel (masking always reduces accuracy). Baselines: Cross-modal: 34.70%, Sensor-only: 35.13%.

| Cross-modal | | Sensor-only | |
|---|---|---|---|
| Channel (gas) | $\Delta$ Acc@1 | Channel (gas) | $\Delta$ Acc@1 |
| $NO_2$ | $-21.27$ | CO | $-25.15$ |
| CO | $-18.80$ | $NO_2$ | $-22.77$ |
| $C_2H_5OH$ | $-18.71$ | $C_2H_5OH$ | $-21.01$ |
| VOC | $-16.15$ | VOC | $-14.03$ |
| Alcohol | $-9.36$ | Alcohol | $-12.44$ |
| LPG | $-8.91$ | LPG | $-10.41$ |

Table 13: Leave-one-day-out performance of SCENTFORMER on SMELLNET-BASE for $p = 25$. We report mean $\pm$ standard deviation over the 6 held-out days for each window size. All values are percentages.

| $w$ | Acc@1 (%) | Acc@5 (%) | F1 (macro, %) |
|---|---|---|---|
| 50 | $49.4 \pm 8.0$ | $84.2 \pm 6.3$ | $48.2 \pm 7.9$ |
| 100 | $53.0 \pm 6.0$ | $85.5 \pm 5.0$ | $51.6 \pm 6.0$ |

## H.4 LEAVE-ONE-DAY-OUT EVALUATION ON SMELLNET-BASE

To assess temporal robustness, we evaluate SCENTFORMER on SMELLNET-BASE using a leave-one-day-out protocol. Each day in SMELLNET-BASE corresponds to a separate acquisition session on a different calendar day with different ambient conditions. In each split, we hold out all samples from one day as the test set and train on the remaining five days. This protocol therefore measures generalization across day-to-day distribution shifts, rather than relying on i.i.d. resampling.

Tab. 13 reports mean $\pm$ standard deviation across the 6 held-out days for $p = 25$ and window sizes $w \in \{50, 100\}$. Tab. 14 provides the per-day breakdown. Across both window sizes, Day 1 is the most challenging split and consistently yields the lowest Acc@1 and macro-F1, suggesting a larger day-specific distribution shift (e.g., ambient conditions or sensor state). Importantly, performance remains well above chance even on the hardest day (2% Acc@1 for 50 classes), and performance on the remaining days clusters substantially higher. Overall, the mean $\pm$ std across days reflects genuine between-day domain differences and indicates that SCENTFORMER generalizes robustly across realistic day-to-day variations in SMELLNET-BASE.

## H.5 TIMESTAMP SIZE ANALYSIS

To justify our 10-minute recording interval, we conducted an ablation that varies the number of initial time steps fed to SCENTFORMER(window size $w=50$) under differencing lags $p \in \{0, 25\}$ (Tab. 15). Using only the first 200 steps yields lower accuracy. As the number of steps increases, performance improves and then plateaus around 400-600 steps. With temporal differencing ($p=25$), Acc@1 increases from 45.7 at 200 steps to 53.2 at 500 steps and remains essentially unchanged at $\approx$600 steps (50.6/52.6 for sensor/cross-modal). This saturation suggests that a 10-minute window captures sufficient temporal dynamics for robust recognition, while longer acquisitions provide diminishing returns. Based on these preliminary findings and practical data-collection constraints, we therefore adopt 10-minute intervals throughout.

## H.6 RUNTIME AND MEMORY ANALYSIS

All experiments use SCENTFORMER on an NVIDIA L40S (compute capability 8.9, driver 550.54.14, CUDA 12.4, 46,068 MiB VRAM) with FP32, batch size = 32, and a ~ 2.4109M-parameter model (live memory ~ 18.9626 MB, peak GPU $\leq$ 75.4966 MB). Inference latency is extremely low across settings: mean per-window latency ranges from 0.0191-0.0479 ms, and throughput remains high at 20,892-52,371 windows per second. As measurements are forward-only, temporal differencing is irrelevant. At this batch size, both the presence of contrastive training and the window size have negligible practical impact on latency.

Table 14: Per-day leave-one-day-out performance of SCENTFORMER on SMELLNET-BASE for $p = 25$ and $w \in \{50, 100\}$. Each row holds out one day as the test set. All values are percentages.

| $w$ | Day | Acc@1 | Acc@5 | F1 (macro) |
|---|---|---|---|---|
| 50 | 1 | 34.8 | 73.5 | 33.3 |
| 50 | 2 | 54.7 | 90.7 | 52.8 |
| 50 | 3 | 55.7 | 89.4 | 54.2 |
| 50 | 4 | 54.5 | 85.9 | 53.2 |
| 50 | 5 | 45.9 | 80.7 | 46.3 |
| 50 | 6 | 50.6 | 85.0 | 49.5 |
| 100 | 1 | 43.6 | 80.8 | 41.4 |
| 100 | 2 | 56.0 | 89.6 | 54.9 |
| 100 | 3 | 57.1 | 89.5 | 54.1 |
| 100 | 4 | 58.0 | 88.0 | 56.6 |
| 100 | 5 | 47.2 | 77.9 | 47.0 |
| 100 | 6 | 56.1 | 87.4 | 55.5 |

Table 15: **Ablation on the number of initial time steps used (SCENTFORMER, window size $w$=50).** $p$ denotes the temporal differencing lag (in samples). Accuracies and F1 are reported in %. Increasing beyond 400-600 steps yields diminishing returns.

| # Timestamps | $p$ | Sensor (SCENTFORMER, $w$=50) | | | Cross-Modal (GC-MS) | | |
|---|---|---|---|---|---|---|---|
| | | Acc@1 | Acc@5 | F1 | Acc@1 | Acc@5 | F1 |
| 200 | 0 | 33.1 | 78.2 | 30.9 | 34.5 | 70.2 | 32.6 |
| 200 | 25 | 45.7 | 80.8 | 43.1 | 35.6 | 66.1 | 30.7 |
| 300 | 0 | 35.2 | 70.1 | 31.1 | 36.2 | 69.4 | 32.8 |
| 300 | 25 | 48.3 | 84.3 | 46.7 | 46.7 | 77.0 | 44.5 |
| 400 | 0 | 35.4 | 73.7 | 32.0 | 37.8 | 68.4 | 33.4 |
| 400 | 25 | 50.5 | 83.3 | 49.2 | 50.5 | 81.8 | 48.6 |
| 500 | 0 | 34.6 | 78.2 | 32.0 | 43.9 | 77.1 | 41.5 |
| 500 | 25 | 53.2 | 87.4 | 51.2 | 50.9 | 83.7 | 49.4 |
| $\approx$600 | 0 | 35.1 | 70.9 | 33.1 | 34.7 | 69.0 | 31.4 |
| $\approx$600 | 25 | 50.6 | 85.0 | 49.5 | 52.6 | 82.5 | 51.7 |

# I  MATH

## I.1  NUMBER OF WINDOWS

Let $T$ be the number of time steps in a recording, $w$ the window length (in steps), and $s$ the stride. Using valid (no-padding) windows, the number of extracted windows is

$$N(T, w, s) = \begin{cases} \left\lfloor \dfrac{T - w}{s} \right\rfloor + 1, & T \geq w, \\ 0, & T < w . \end{cases} \tag{1}$$

We use $50\%$ overlap, i.e., $s = \frac{w}{2}$. Thus a 10-minute recording at 1 Hz has $T = 600$ steps and yields

$$N(600, 50, 25) = \left\lfloor \frac{600 - 50}{25} \right\rfloor + 1 = 22 + 1 = 23, \quad N(600, 100, 50) = \left\lfloor \frac{600 - 100}{50} \right\rfloor + 1 = 10 + 1 = 11.$$

With sampling rate $f_s$ (Hz) and duration $L$ (s), $T = f_s L$, and each window spans $w/f_s$ seconds. If right-padding is used to include a final partial window, replace the floor in (1) with a ceiling.

## I.2  TOP-K PERFORMANCE CALCULATION

For each example $n \in \{1, \dots, N\}$ with predicted class probabilities $\boldsymbol{p}^{(n)} = \mathrm{softmax}(\boldsymbol{s}^{(n)}) \in [0, 1]^C$, let $R_n = \{c : y_c^{(n)} > 0\}$ denote the nonempty set of ground-truth present classes and $P_n = |R_n|$ its cardinality. Let $\pi_n(1), \dots, \pi_n(C)$ index classes in descending order of $\boldsymbol{p}^{(n)}$, and let $\Pi_n(k) = \{\pi_n(1), \dots, \pi_n(k)\}$ denote the set of top-$k$ predicted classes. The dynamic Top-$K$ metric is defined

Table 16: Latency and resource metrics for **SCENTFORMER** (batch=32, FP32) on NVIDIA L40S.

| Variant | Win | Mean (ms) | WPS | Eval (ms) | GPU MB | Live MB | Params (M) |
|---|---|---|---|---|---|---|---|
| No contrastive | 50 | 0.0191 | 52371 | 0.0189 | 63.3540 | 18.9626 | 2.4109 |
| Contrastive | 50 | 0.0203 | 49237 | 0.0310 | 61.0024 | 18.9626 | 2.4109 |
| No contrastive | 100 | 0.0479 | 20892 | 0.0218 | 73.6172 | 18.9626 | 2.4109 |
| Contrastive | 100 | 0.0420 | 23821 | 0.0151 | 75.4966 | 18.9626 | 2.4109 |

as the label recall under a per-example cutoff $K_n = P_n$:

$$\text{DynTopK} = \frac{\sum_{n=1}^{N} |R_n \cap \Pi_n(P_n)|}{\sum_{n=1}^{N} |R_n|} \times 100\%. \tag{2}$$

In the SMELLNET-MIXTURE evaluation, $N = 12$ odorant classes are considered. Defining $H = \sum_{n=1}^{12} |R_n \cap \Pi_n(P_n)|$ and $M = \sum_{n=1}^{12} |R_n|$, the metric reduces to $\text{DynTopK} = (H/M) \times 100\%$. In the special case where each example contains exactly one present class (i.e., $|R_n| = 1$ for all $n$), the denominator simplifies to $M = N$, and DynTopK is equivalent to standard Top-1 accuracy over the $N$ examples.

## J  DETAILS OF SMELL MIXTURES

We define 43 unique ingredient combinations, and instantiate 126 unique recipes by assigning multiple ratios to each combination. For the collection of mixture data, each base odorant was treated as a distinct class, yielding 12 base classes. Beyond these bases, we constructed binary and ternary mixtures of the base odorants at fixed volumetric ratios, resulting in a total of 126 unique mixture combinations (54 in training, 45 in test-seen, 27 in test-unseen). The dataset exhibits the following mixture distribution:

- **Base odorants**: 24.7% of training sessions (168 sessions), 22.3% of test-seen (48 sessions), 0% of test-unseen

- **Binary mixtures**: 71.3% of training sessions (484 sessions), 73.5% of test-seen (158 sessions), 83.2% of test-unseen (153 sessions)

- **Ternary mixtures**: 4.0% of training sessions (27 sessions), 4.2% of test-seen (9 sessions), 16.8% of test-unseen (31 sessions)

The composition of odor readings is shown in Fig. 3. Binary mixture ratios in the training set span multiple combinations including 20/80 (160 sessions), 50/50 (174 sessions), and 80/20 (120 sessions), with additional ratios spanning from 10/90 to 90/10. Ternary mixtures include both balanced (33/33/33) and asymmetric (10/30/60) distributions. This comprehensive ratio coverage enables the model to learn ratio prediction across the full spectrum of possible combinations.

## K  LLM USAGE

We disclose a limited use of a large language model (LLM) during development and manuscript preparation.

**Manuscript editing.**  We used an LLM for light copy-editing of the paper text (e.g., grammar, phrasing, and clarity improvements). All technical content, experiments, analyses, figures, tables, and claims were written, checked, and approved by the authors.

**Limited technical use in an early prototype.**  During an early prototype of the *element-count* GC-MS descriptor pipeline (an alternative encoding reported for comparison in App. C), some FooDB entries did not include chemical formulas. In a small number of such cases, we queried an LLM once to *suggest* a candidate molecular formula. These suggestions were *not* used directly. Each chemical formula was manually verified against public chemistry databases and our own parsing and validation code before inclusion.

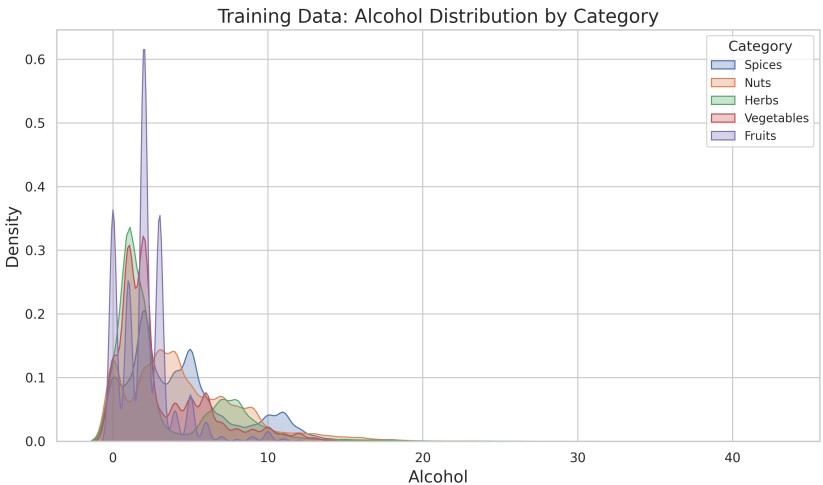

Figure 16: **KDE distribution of Alcohol Sensor by Category.** These numbers represent raw, unfiltered data readings. Normalizations and filtering of abnormal channels are performed as preprocessing steps before modeling. We provide standard kits for preprocessing, but the raw values are provided to preserve as much information as possible.

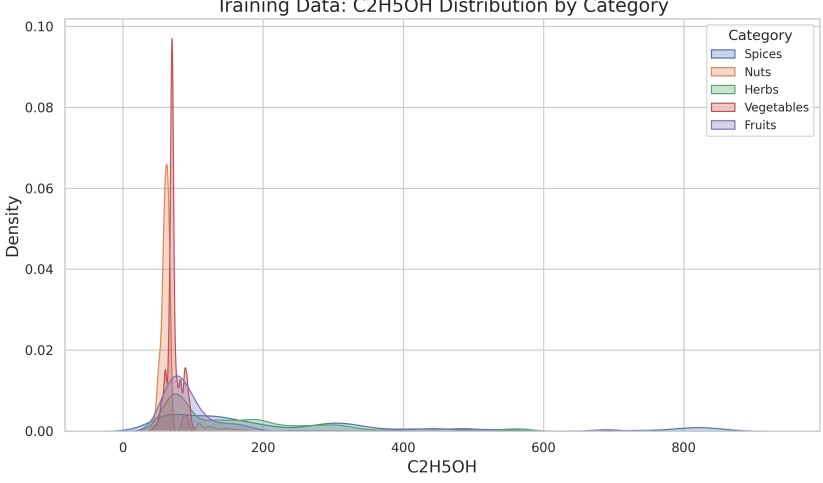

Figure 17: **KDE distribution of C2H5OH Sensor by Category.**

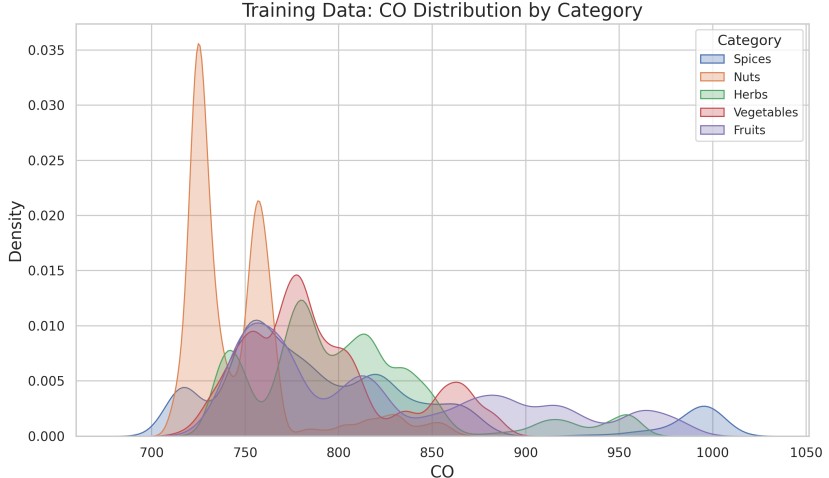

Figure 18: **KDE distribution of CO Sensor by Category.**

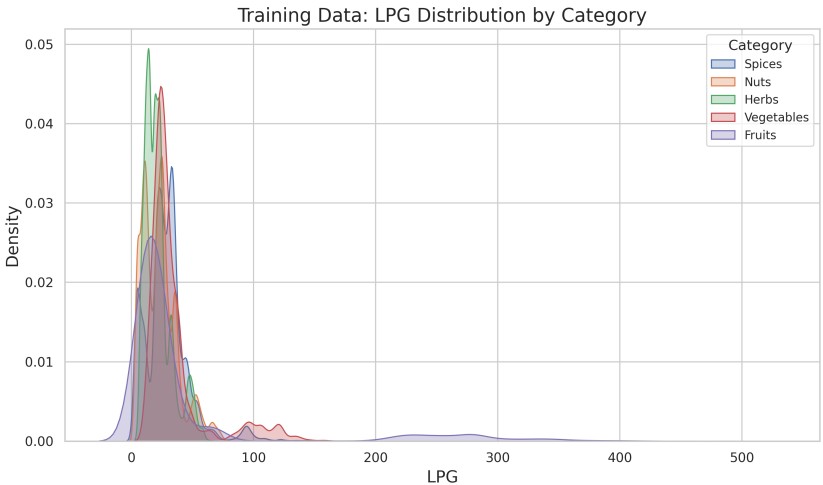

Figure 19: **KDE distribution of LPG Sensor by Category.**

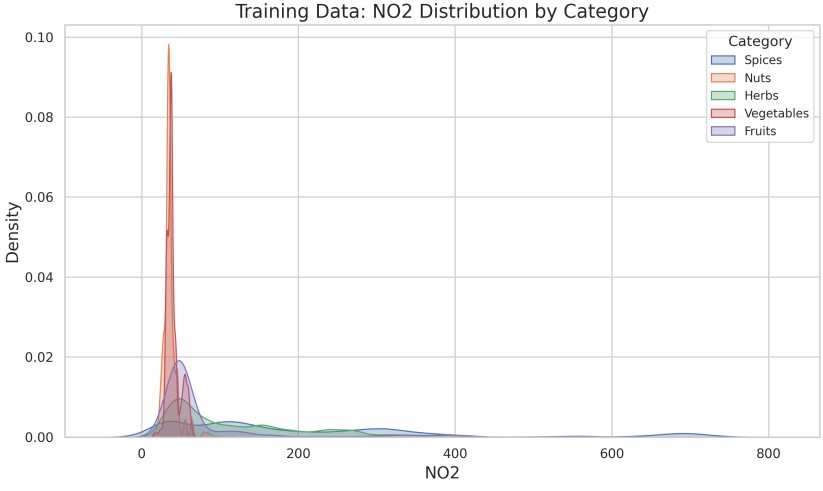

Figure 20: **KDE distribution of NO2 Sensor by Category.**

Table 17: Olfactory materials used in SMELLNET, organized alphabetically by label. We distinguish essential oils (volatile natural extracts with identifiable key odorants), flavor extracts (culinary preparations in oil or alcohol carriers), fragrance oils (synthetic or proprietary blends), and cosmetic oils (carrier oils with an odor).

| Label | Type | Material | Vendor |
|---|---|---|---|
| Almond | Flavor Extract | Pure Almond Extract | CADIA |
| Apple | Fragrance Oil | Apple Fragrance Oil | P&J Trading |
| Banana | Cosmetic Oil | Banana Oil | The Aromatherapy Shop Ltd |
| Clove | Essential Oil | Clove Bud Essential Oil | 365 Whole Foods Market |
| Coriander | Essential Oil | Coriander Essential Oil | Skylara Essentials |
| Cumin | Essential Oil | Cumin Essential Oil | Silky Scents |
| Garlic | Essential Oil | Garlic Essential Oil | Skylara Essentials |
| Mango | Essential Oil | Mango Essential Oil (Egypt) | The Aromatherapy Shop Ltd |
| Orange | Flavor Extract | Orange Flavor | Frontier Co-op Store |
| Peach | Fragrance Oil | Peach Fragrance Oil | P&J Trading |
| Pear | Fragrance Oil | Pear Fragrance Oil | P&J Trading |
| Strawberry | Fragrance Oil | Strawberry Fragrance Oil | P&J Trading |

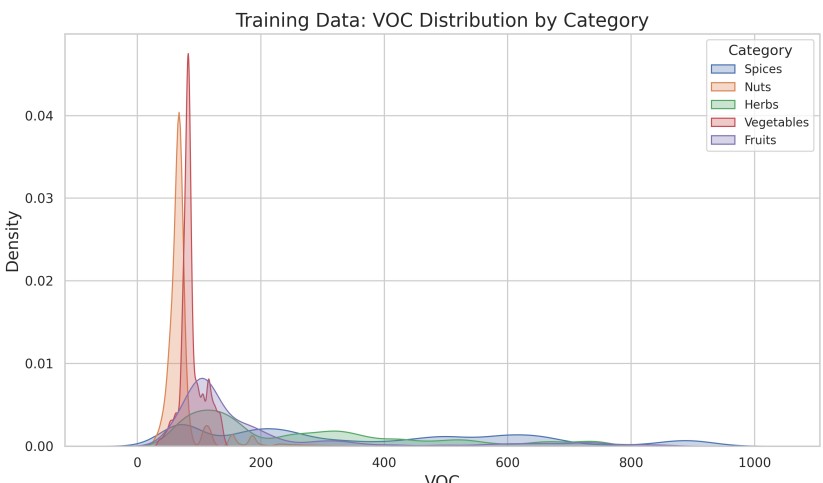

Figure 21: **KDE distribution of VOC Sensor by Category.**

**No LLM use in model development or evaluation.** We did not use an LLM to design experiments, select models or hyperparameters, generate results for model training or evaluation, analyze outcomes, or produce scientific conclusions. No LLM outputs were used as training data, labels, or inputs to the reported sensor models.

**Reproducibility.** To ensure reproducibility without any LLM dependency, we release the final deterministic scripts used to construct the GC-MS descriptors, together with the finalized compound-formula table and processed GC-MS embeddings. Reproducing the reported results does not require calling any LLM.

Table 18: **Statistics of Gas Sensor Measurements** These numbers represent raw, unfiltered data readings. Normalizations and filtering of abnormal channels are performed as preprocessing steps before modeling. We provide standard kits for preprocessing, but the raw values are provided to preserve as much information as possible.

| Ingredient | NO2 mean | NO2 std | C2H5OH mean | C2H5OH std | VOC mean | VOC std | CO mean | CO std | Alcohol mean | Alcohol std | LPG mean | LPG std |
|---|---|---|---|---|---|---|---|---|---|---|---|---|
| allspice | 257.35 | 47.64 | 298.64 | 46.79 | 493.35 | 72.72 | 835.91 | 23.98 | 2.42 | 0.52 | 34.23 | 1.75 |
| almond | 58.10 | 15.99 | 112.52 | 26.03 | 164.75 | 61.27 | 735.84 | 12.07 | 4.38 | 0.66 | 20.64 | 1.09 |
| angelica | 143.16 | 41.41 | 175.46 | 32.63 | 316.82 | 69.57 | 769.97 | 9.79 | 1.07 | 0.73 | 8.91 | 1.89 |
| apple | 45.38 | 3.91 | 66.93 | 2.40 | 126.27 | 28.85 | 750.08 | 7.97 | 2.25 | 0.44 | 10.58 | 0.53 |
| asparagus | 54.49 | 6.42 | 87.34 | 7.87 | 117.50 | 15.95 | 780.18 | 14.74 | 4.26 | 1.84 | 18.93 | 3.44 |
| avocado | 31.56 | 1.11 | 69.69 | 1.31 | 77.40 | 3.22 | 860.57 | 14.95 | 1.72 | 0.45 | 24.55 | 3.32 |
| banana | 58.15 | 7.60 | 89.30 | 7.99 | 154.88 | 34.11 | 804.62 | 18.02 | 2.71 | 0.52 | 31.82 | 3.16 |
| brazil nut | 32.12 | 3.84 | 58.70 | 3.63 | 61.27 | 8.54 | 727.97 | 5.98 | 2.42 | 0.59 | 23.91 | 1.10 |
| broccoli | 35.14 | 3.87 | 69.14 | 5.73 | 72.25 | 11.89 | 855.23 | 22.45 | 1.05 | 0.23 | 34.33 | 4.26 |
| brussel sprouts | 42.56 | 5.18 | 65.20 | 3.23 | 84.28 | 8.67 | 744.62 | 9.68 | 3.73 | 2.66 | 20.03 | 3.83 |
| cabbage | 37.25 | 1.86 | 70.06 | 1.48 | 82.76 | 3.74 | 776.35 | 6.04 | 1.67 | 0.48 | 25.44 | 1.11 |
| cashew | 34.52 | 1.28 | 66.05 | 0.84 | 67.58 | 2.70 | 757.08 | 1.83 | 5.59 | 2.94 | 13.81 | 2.51 |
| cauliflower | 36.75 | 1.08 | 69.59 | 0.90 | 81.63 | 1.98 | 777.96 | 6.62 | 1.29 | 0.46 | 30.06 | 1.66 |
| chamomile | 47.79 | 5.60 | 69.86 | 7.89 | 104.80 | 14.10 | 752.26 | 19.14 | 1.41 | 0.73 | 12.40 | 1.73 |
| chervil | 38.36 | 3.65 | 62.39 | 1.86 | 77.55 | 5.74 | 718.41 | 7.32 | 5.65 | 2.72 | 33.60 | 5.43 |
| chestnuts | 34.51 | 5.07 | 60.54 | 2.88 | 67.40 | 8.48 | 819.79 | 26.96 | 11.49 | 3.23 | 54.55 | 9.19 |
| chives | 46.26 | 4.93 | 80.56 | 6.78 | 114.11 | 16.46 | 801.50 | 11.77 | 7.37 | 1.23 | 22.74 | 2.96 |
| cinnamon | 163.05 | 37.50 | 182.66 | 31.10 | 374.53 | 96.46 | 787.42 | 15.21 | 3.20 | 1.01 | 23.16 | 2.55 |
| cloves | 121.25 | 34.45 | 105.63 | 25.27 | 238.72 | 69.35 | 778.28 | 7.02 | 11.29 | 1.66 | 43.08 | 4.02 |
| coriander | 237.54 | 38.93 | 275.84 | 53.13 | 481.83 | 70.99 | 830.32 | 16.57 | 1.42 | 0.52 | 23.82 | 4.30 |
| cumin | 322.36 | 64.14 | 454.09 | 62.83 | 608.59 | 75.88 | 848.05 | 16.92 | 5.22 | 1.63 | 31.79 | 3.09 |
| dill | 305.26 | 75.62 | 472.43 | 85.59 | 660.98 | 105.97 | 927.31 | 29.28 | 9.29 | 4.52 | 33.76 | 5.21 |
| garlic | 75.73 | 11.98 | 111.27 | 23.07 | 149.23 | 32.19 | 826.53 | 26.66 | 1.36 | 0.73 | 15.95 | 1.90 |
| ginger | 281.14 | 53.82 | 281.85 | 45.31 | 585.09 | 90.61 | 810.70 | 17.01 | 5.19 | 0.50 | 20.41 | 0.76 |
| hazelnut | 35.27 | 1.92 | 62.76 | 0.71 | 71.75 | 2.57 | 727.42 | 2.93 | 3.62 | 1.15 | 26.62 | 1.85 |
| kiwi | 30.99 | 2.33 | 63.09 | 2.98 | 63.09 | 6.45 | 771.93 | 2.86 | 1.02 | 0.15 | 20.80 | 1.12 |
| lemon | 373.19 | 108.93 | 497.90 | 109.86 | 672.55 | 120.62 | 877.10 | 44.48 | 2.89 | 0.50 | 14.61 | 3.64 |
| mandarin | 111.15 | 35.02 | 145.58 | 33.67 | 246.67 | 74.52 | 872.39 | 35.47 | 1.59 | 0.53 | 23.36 | 3.30 |
| mango | 49.58 | 6.39 | 82.23 | 8.90 | 115.03 | 13.03 | 757.49 | 7.69 | 2.04 | 0.22 | 11.91 | 0.86 |
| mint | 60.52 | 4.89 | 81.03 | 6.27 | 141.44 | 17.75 | 745.44 | 4.56 | 0.17 | 0.38 | 12.38 | 1.36 |
| mugwort | 116.18 | 19.94 | 171.32 | 33.38 | 264.83 | 41.12 | 786.64 | 6.50 | 1.19 | 0.43 | 18.76 | 1.02 |
| mustard | 29.07 | 3.12 | 65.48 | 3.93 | 63.17 | 8.33 | 754.74 | 3.97 | 1.22 | 0.44 | 10.43 | 1.88 |
| nutmeg | 647.16 | 108.49 | 789.17 | 94.66 | 850.61 | 99.73 | 985.72 | 29.25 | 6.76 | 1.80 | 66.55 | 25.78 |
| oregano | 176.32 | 33.02 | 245.84 | 51.77 | 344.29 | 48.64 | 794.02 | 14.98 | 7.09 | 3.08 | 45.97 | 7.72 |
| peach | 64.74 | 23.37 | 153.68 | 67.61 | 224.51 | 86.09 | 956.07 | 27.04 | 5.98 | 2.01 | 262.50 | 63.55 |
| peanuts | 35.84 | 2.00 | 64.09 | 2.16 | 63.25 | 5.72 | 753.14 | 2.78 | 1.60 | 1.18 | 8.60 | 1.62 |
| pear | 40.39 | 3.34 | 67.87 | 2.88 | 98.02 | 9.45 | 802.24 | 16.63 | 2.32 | 0.48 | 18.23 | 2.90 |
| pecans | 25.14 | 3.14 | 49.67 | 3.58 | 47.55 | 8.22 | 726.83 | 4.23 | 7.28 | 1.64 | 30.81 | 3.52 |
| pili nut | 33.74 | 1.63 | 57.68 | 0.52 | 67.48 | 1.88 | 726.42 | 3.57 | 7.07 | 1.58 | 36.68 | 2.01 |
| pineapple | 47.09 | 5.87 | 82.72 | 6.47 | 102.16 | 12.14 | 897.35 | 26.27 | 0.00 | 0.00 | 48.32 | 19.73 |
| pistachios | 39.79 | 4.90 | 56.79 | 3.38 | 69.73 | 9.04 | 720.86 | 6.71 | 0.00 | 0.00 | 4.78 | 0.81 |
| potato | 32.79 | 5.49 | 56.15 | 4.86 | 65.47 | 14.22 | 745.50 | 11.72 | 5.53 | 1.48 | 40.54 | 4.37 |
| radish | 46.52 | 8.50 | 82.89 | 9.08 | 102.48 | 17.90 | 787.99 | 21.02 | 0.00 | 0.00 | 32.07 | 17.94 |
| saffron | 110.59 | 13.60 | 150.79 | 12.46 | 220.78 | 22.85 | 763.06 | 4.90 | 1.99 | 0.12 | 25.14 | 2.42 |
| star anise | 91.89 | 21.29 | 115.33 | 17.30 | 173.25 | 33.78 | 744.94 | 6.98 | 0.00 | 0.00 | 4.47 | 0.60 |
| strawberry | 54.26 | 6.89 | 91.86 | 10.47 | 108.25 | 12.18 | 749.20 | 10.58 | 0.00 | 0.00 | 3.67 | 0.50 |
| sweet potato | 38.58 | 2.85 | 73.99 | 4.61 | 98.93 | 9.31 | 763.77 | 11.52 | 8.83 | 2.51 | 107.87 | 16.80 |
| tomato | 39.76 | 10.09 | 73.82 | 6.74 | 90.18 | 19.60 | 809.73 | 13.25 | 1.60 | 0.49 | 22.61 | 1.92 |
| turnip | 30.31 | 0.79 | 67.94 | 1.42 | 71.80 | 2.71 | 821.85 | 9.80 | 2.00 | 0.75 | 22.17 | 1.69 |
| walnuts | 32.67 | 5.04 | 63.04 | 3.21 | 59.40 | 8.73 | 762.96 | 2.53 | 2.38 | 1.89 | 12.39 | 2.28 |

