# OpenReview forum: "SmellNet: A Large-scale Dataset for Real-world Smell Recognition"
_ICLR.cc/2026/Conference — ICLR 2026 Poster_

### Official Review · Reviewer_wU3A · 2025-10-27

**Soundness:** 2
**Presentation:** 4
**Contribution:** 3
**Rating:** 4
**Confidence:** 4

**Summary:**

In this paper the authors contribute SmellNet, a novel large-scale dataset of chemical data for olfactory recognition tasks. In particular, the authors collect over 68 hours of gas sensor data, pertaining to 50 different substance and 43 artificial mixtures of compounds. Furthermore, the authors present SmellFormer, a transformer-based architecture for odor prediction from sensor data. Finally, the authors extensively evaluate ScentFormer on single-component and mixture odor recognition data, highlighting how their transformer architecture outperforms other standard architectures (MLP, CNN, LSTM) and the potential of their dataset for future applications in olfactory machine learning.

**Strengths:**

- **Originality**: While large-scale datasets of olfactory phenomena exist and have been extensively used by the community for olfactory prediction tasks ([1, 2]), the dataset presented here distinguishes itself by using directly sensor readings, collected using consumer-available hardware (instead of GC-MS and other specialized equipment).

- **Quality**: I found the paper to be of high quality, with minimal typos, high-quality figures and text. The authors extensively evaluate the discriminative power of the collected dataset, both in the main text and in appendix. Furthermore, the authors evaluate different experimental setups, in terms of preprocessing steps, additional modalities and model architectures, for single and mixture odor prediction.

- **Clarity**: Overall, the authors describe in extensive detail their data collection setup (hardware used and processing pipeline) and model architecture (preprocessing steps, architecture and training objectives). I also deeply appreciated the structure of the evaluation section, with clear research questions and highlighting the main findings for each of them.

- **Significance**: The lack of large-scale, diverse, olfactory data, collected in-the-wild, is currently a major bottleneck for the olfactory machine learning community. While being part of ongoing data collection efforts by the community, the use of consumer-available hardware to collect SmellNet makes this work a potential interesting resource for the community.

**References**:

- [1] Lee, Brian K., et al. "A principal odor map unifies diverse tasks in olfactory perception." Science 381.6661 (2023): 999-1006.
- [2] Taleb, Farzaneh, et al. "Can transformers smell like humans?." Advances in Neural Information Processing Systems 37 (2024): 72032-72060.

**Weaknesses:**

- I would suggest the authors to refrain from over claiming in certain statements. For example, statements such as "smell is a new data modality for AI (line 51)", or "large-scale AI for smell is completely unexplored (line 91)" do not only overlooks important advances from the chemical and neuroscience community (some of them, already cited in the paper) on using AI for olfactory prediction tasks, but it is also factually incorrect: "large-scale AI" (whatever that means) have been used for olfactory prediction tasks, see [1], [2], for example.

- I found it quite interesting that throughout the work the authors state that they collect sensory gas data in 12-channels, and evaluate their dataset under these conditions (Section 3.4), yet the first step in data preprocessing (Section 4.2) is to drop 6 of those channels, due to potential malfunctioning of these sensors. From a practical point of view, the authors should only release the 6-channel dataset (which I don't think would significantly decrease the novelty or significance of the work), or redo the data collection with functioning sensors. Releasing the full 12-channel dataset, where 6 of those are malfunctioning, can lead to future issues when people may employ the dataset without removing the channels. Is there any reason to release the full dataset, beyond the ones stated in Line 268-270, which are not particularly beneficial considering the previously discussed danger?

- While the authors describe extensively the statistics and experimental apparatus to collect SmellNet, I found the description of the GC-MS pairings (which, accordingly to the results in Section 5, improves the performance of the predictive models) to be lacking and confusing. GC-MS is a high-precision measurement of the compounds present in a sample and, as such, varies across samples. Moreover, to the best of my knowledge, there exists no single GC-MS measurement/spectrum for some of the samples employed in your dataset (e.g., Apple, Pineapple). How did the authors collect this data? Moreover, how did the authors take the spectrum data and get the atomic counts (which is an unusual representation of GC-MS data, what about concentration for example)? The Appendix C mentions the use of an LLM for this purpose, yet it's unclear what is the process. Also, this use of LLMs for technical content goes against what is claimed in Appendix K, regarding the use of LLMs, which the authors to be used "solely for light copy-editing".

**References**:

- [1] Lee, Brian K., et al. "A principal odor map unifies diverse tasks in olfactory perception." Science 381.6661 (2023): 999-1006.
- [2] Taleb, Farzaneh, et al. "Can transformers smell like humans?." Advances in Neural Information Processing Systems 37 (2024): 72032-72060.

**Questions:**

See Weaknesses.

---

> ### Author Response · Authors · 2025-11-21
> **Author's Response - Thank you for your feedback (part 1)**
>
> We thank the reviewer for the thoughtful and constructive feedback and for finding our work innovative and interesting. We address the main concerns below.
>
> We agree that some of our statements in the introduction were overly strong in light of recent large-scale olfactory work such as principal-odor-map models [1] and transformer-based olfactory prediction [2]. In the revised paper we will
> – remove absolute claims such as “smell is a new data modality for AI” and “large-scale AI for smell is completely unexplored”, and
> – explicitly acknowledge existing large-scale molecular and psychophysical olfactory datasets and models [1, 2].
>
> Instead, we will position our contribution more narrowly: compared to vision and language, olfactory datasets and models remain relatively scarce for sensor-based time-series data collected with portable, low-cost hardware. SmellNet is complementary to principal-odor-map and molecular-descriptor work: we focus on gas sensors in physical environments, mixtures, and sensor–GC-MS alignment, which are currently underexplored. In the camera-ready we will also revise Table 1 / related work to situate SmellNet alongside both human-perception datasets (Dravnieks, DREAM, POM, etc.) and existing e-nose datasets (e.g., coffee, beef, fruit-freshness, and gas-sensor drift benchmarks [1, 2, 3, 4, 5]), making clear that SmellNet is intended as a benchmark for this hardware-proximal layer of olfactory AI rather than as a replacement for perception-level datasets.
>
> > 12-channel hardware vs 6 usable channels and dataset release
>
> We agree that our current description of the 12-channel board can be confusing. The physical hardware exposes 12 channels, but during collection we found that 6 of the gas channels were unreliable on a subset of the recordings. All experiments in the paper are conducted using only the 6 reliable channels.
> To avoid misuse while maintaining transparency, we will:
> 1. Present SmellNet as a 6-channel gas-sensor dataset for modeling purposes, and clearly state in Se. 3.4 and 4.2 that 6 channels are dropped due to reliability issues (updating text, tables, and figures accordingly).
>
> 2. Release two dataset versions:
> - a processed 6-channel version (with only the functioning channels and full documentation)
>
> - a raw 12-channel version in a separate folder, clearly marked as containing malfunctioning channels for archival purposes only.
>
> This preserves transparency while making the recommended 6-channel usage straightforward for future users.
>
> > GC-MS pairing, representation, and conclusions
>
> **Role of GC–MS**
>
> We appreciate the request for a clearer explanation. GC–MS is used only as ingredient level label supervision, not as a per-sample input modality. For each ingredient that appears in FooDB with annotated volatile compounds and electron-ionization mass spectra, we precompute a single GC-MS embedding from FooDB (we do not run GC-MS on our own samples). These embeddings should be viewed as canonical, ingredient-level priors that summarize typical volatile composition, and they are not meant to be exact measurements of our specific physical samples. During training, we apply a CLIP style contrastive loss that pulls sensor windows of that ingredient toward its GC-MS embedding and pushes them away from others. At test time, we use only gas sensor data, and GC-MS is required only to precompute one label embedding per ingredient.
>
> **Element-count GC-MS embedding (original method).**
>
> For the original element count representation, we:
> 1. Query FooDB for each ingredient to obtain its annotated volatile compounds and keep the top 10 ranked by reported content/mass.
> 2. For each compound, obtain a chemical formula (using FooDB when available, and in early prototypes we used an LLM once to suggest a formula when missing, then verified it with public chemistry databases and our own parsing code).
> 3. Convert each formula into an element-count vector, weight these vectors by compound abundance, and sum to obtain a single ingredient level atomic profile.
>
> This vector is used only as a label embedding in the contrastive loss, and it is never used as a per-sample input.
>
> **Strength of GC–MS effect.**
>
> We agree that our wording in Findings 2.1 and 2.2 overstated the effect. As Table 2 shows, GC-MS supervision gives clear but modest gains for the weaker MLP model, and only small, sometimes mixed changes (−1.8 to +4.9 Acc@1) for the stronger temporal models (CNN/LSTM/ScentFormer), which already capture much of the discriminative structure from the time series alone. In the revision we will soften these statements accordingly (like “modest gains for weaker models, and small and mixed effects for stronger temporal models”).

---

> ### Author Response · Authors · 2025-11-21
> **Author's Response - Thank you for your feedback (part 2)**
>
> (continued author's response)
>
> >GC-MS pairing, representation, and conclusions
>
> **New spectral GC-MS representation.**
>
> To address the concern about the element count encoding, we additionally introduce a spectral GC-MS representation based on raw spectra. For each ingredient we:
> 1. Collect its annotated volatile organic compounds from FooDB
> 2. Retrieve all available experimental EI mass spectra for those compounds
> 3. Convert each spectrum into a binned intensity vector over 40–500 m/z (1 Da bins, max-normalized), then average across spectra and compounds to obtain a fixed-length spectral embedding
>
> The 40–500 m/z range covers 99% of the intensity mass in the FooDB spectra we use. On the sensor side, the encoder, training schedule, and evaluation protocol are unchanged. Only the GC-MS label embedding is swapped (element counts to spectra).
>
> **Table 1.** Comparison of single-ingredient classification on SmellNet-Base with three GC-MS settings: sensor-only baseline (no GC-MS, labeled S), cross-modal training with element count GC-MS embeddings ($X_{atom}$), and cross-modal training with spectral GC-MS embeddings ($X_{spec}$). We report Top-1 accuracy (Acc@1, %) for each model, window size w, and temporal gap g. $Δ_{atom}$ and $Δ_{spec}$ denote the change in Acc@1 relative to the sensor-only baseline. Spectral GC-MS supervision shows the same qualitative behavior as the original element-count encoding, where largest gains for MLPs and small, mixed effects for stronger temporal models.
>
> | Model       | w   | g   | S | $X_{atom}$ | $X_{spec}^*$ | $Δ_{atom}$ | $Δ_{spec}^*$ |
> |------------|-----|-----|-----------------|--------|--------|--------|--------|
> | MLP        | 50  | 0   | 21.9| 23.6   | 24.1   | +1.7   | +2.2   |
> | MLP        | 50  | 25  | 18.2| 23.8   | 24.7   | +5.6   | +6.5   |
> | MLP        | 100 | 0   | 21.0| 25.6   | 24.3   | +4.6   | +3.3   |
> | MLP        | 100 | 25  | 26.8| 28.0   | 28.5   | +1.2   | +1.7   |
> | CNN        | 50  | 0   | 25.5| 28.4   | 27.7   | +2.9   | +2.2   |
> | CNN        | 50  | 25  | 46.9| 45.9   | 45.7   | −1.0   | −1.2   |
> | CNN        | 100 | 0   | 29.5| 31.0   | 26.7   | +1.5   | −2.8   |
> | CNN        | 100 | 25  | 52.7| 57.1   | 55.0   | +4.4   | +2.3   |
> | LSTM       | 50  | 0   | 29.3| 29.7   | 31.9   | +0.4   | +2.6   |
> | LSTM       | 50  | 25  | 50.6| 53.3   | 50.7   | +2.7   | +0.1   |
> | LSTM       | 100 | 0   | 28.8| 33.7   | 35.2   | +4.9   | +6.4   |
> | LSTM       | 100 | 25  | 57.9| 56.1   | 56.2   | −1.8   | −1.7   |
> | Transformer| 50  | 0   | 35.1| 36.2   | 34.8   | +1.1   | −0.3   |
> | Transformer| 50  | 25  | 50.6| 50.9   | 52.6   | +0.3   | +2.0   |
> | Transformer| 100 | 0   | 39.9| 41.0   | 39.5   | +1.1   | −0.4   |
> | Transformer| 100 | 25  | 56.1| 58.5   | 55.8   | +2.4   | −0.3   |
>
> \* means new to this rebuttal
>
> Taken together, these results show that our RQ2 conclusions are robust to the choice of GC-MS encoding. Whether we use an element count vector or a full spectral representation, GC-MS functions as an optional chemistry aware prior that modestly regularizes the sensor embedding, and the main performance gains still come from modeling the sensor time series.
>
> We will release all GC-MS related artifacts, including the generated compound formula files and the processed GC-MS embeddings (atomic and spectral), as part of the public dataset.
>
> > LLM usage clarification
>
> We appreciate the comment about our use of LLMs. Our intent in Appendix K was to indicate that we did not use LLMs to design experiments, choose models, or automatically generate technical methods. In practice, we used an LLM in one narrow way during an early prototype: when FooDB did not list a chemical formula for a compound, we queried an LLM once to propose a formula and then manually verified it against public chemistry databases and our own parsing code. We also used an LLM for light copy-editing of the text. In the camera ready version, we will revise Appendix K to explicitly describe this limited technical use, and we will release the final deterministic scripts and the compound formula table so that reproducing our GC-MS embeddings does not require any LLM.
>
> **References**
>
> [1] Rodríguez, Juan, et al. "Electronic nose for quality control of Colombian coffee through the detection of defects in 'Cup Tests'." Sensors 10.1 (2010): 36-46.
>
> [2] Wijaya, Dedy Rahman, Riyanarto Sarno, and Enny Zulaika. "Electronic nose dataset for beef quality monitoring in uncontrolled ambient conditions." Data in Brief 21 (2018): 2414-2420.
>
> [3] Mahradian, Mehrab. "Fruit Freshness, Electronic Nose Dataset." Kaggle dataset (2023).
>
> [4] Fonollosa, Jordi, Irene Rodríguez-Luján, and Ramón Huerta. "Chemical gas sensor array dataset." Data in Brief 3 (2015): 85-89.
>
> [5] Vergara, Alexander. "Gas Sensor Array Drift Dataset at Different Concentrations." UCI Machine Learning Repository (2013).

---

> > ### Comment · Reviewer_wU3A · 2025-11-24
> > **Response to Author's Rebuttal**
> >
> > I thank the authors for their detailed rebuttal that addressed some of my concerns. I have, however, some remaining issues regarding the experimental results.
> >
> > In particular, the authors state that "All experiments in the paper are conducted using only the 6 reliable channels". However, this appears to be inconsistent with multiple parts of the manuscript: the results presented in Appendix B1 and B2 clearly employ the full set of 12-channel features; the results in Table 15-16, and Figures 13-22, also employ the 12-channel features.
> >
> > In addition to this uncertainty about which set of features is used in which part of the paper, the decision to release the dataset with the 12-channel features (and claim it as a contribution!), already knowing that it includes erroneous values for half of these features, raises significant methodological concerns.
> >
> > To avoid this issue, the authors should either (i) release only the corrected 6-channel dataset and revise the paper accordingly, or (ii) repeat the data collection to ensure all 12 channels are valid and re-run the experiments.

---

> > > ### Author Response · Authors · 2025-11-25
> > > **Thank you for your feedback!**
> > >
> > > We thank the reviewer for pointing out the inconsistency between our rebuttal statement and some parts of the appendix. While we believe that the remaining 6 channels with partially invalid data still offer research values, we agree that this data is out of scope for this work, and the current models are not able to make effective use of the data. As a result, we will remove the parts about 12 channels and mention 6 channels for now.
> > >
> > > For the camera-ready version we will take the following steps:
> > >
> > > We will explicitly define SmellNet-Base as a 6-channel benchmark dataset, and state in Sections 3.4 and 4.2 that all reported benchmark results, including the new ablations from the rebuttal, use only these 6 functioning channels.
> > > We will remove remaining 12-channel analyses in the appendix. Any plots/tables we keep will be updated to use the 6-channel features, and we will add a clear note at the start of the appendix that all benchmark results use 6 channels.
> > > The primary public release will be a cleaned 6-channel dataset with full documentation, which is what we intend other researchers to use for modeling and comparison.
> > > Our goal is to be conservative and transparent. The accurate 6-channel dataset will be the only one used for benchmarks and highlighted as the main contribution, while the raw 12-channel logs, if released at all, will be clearly documented as archival data that may contain unreliable sensors.
> > >
> > > We appreciate the reviewer’s insistence on this point! It has helped us tighten both the dataset definition and the presentation.

---

> > > > ### Comment · Reviewer_wU3A · 2025-11-26
> > > > **Response to Author's Rebuttal - 2**
> > > >
> > > > I thank the authors for their clarification, and for their promise to update the document with these changes. Given this, I increase my score to 6.

---

> > > > > ### Author Response · Authors · 2025-12-03
> > > > > **Thanks for raising the score**
> > > > >
> > > > > We sincerely thank reviewer wU3A for actively engaging in our rebuttal and raising the score to 6. We will incorporate all feedbacks in our final manuscript.

---

### Official Review · Reviewer_eaQT · 2025-10-29

**Soundness:** 3
**Presentation:** 3
**Contribution:** 3
**Rating:** 8
**Confidence:** 4

**Summary:**

This paper introduces a new olfactory dataset, combining temporal sensor data with with object labels, and proposes several models to decode single- and multi-source odorants to establish the baseline results for these data.

**Strengths:**

- A new dataset

The paper offers a new large-scale dataset connecting the odorants and the objects emitting them. The dataset has a reasonable number of odorant (50; typical for olfaction; compare to ~100 in the Leffingwell dataset) and a large number of data point (recorded by the sensors), thus enabling the application of larger machine-learning models than previously (e.g., GNNs and CNNs).

- Complementary approach to olfactory data

While conventionally the olfactory datasets were focused on psychophysical processing of the smell (exposing human participants or animals to smells and recording their semantic or similarity reports respectively), this dataset fully avoids the perceptual properties of the smell, going directly from sensor readouts to object classification. While this choice shifts the scope of what's can be learned from the olfactory dataset, it also enables the large-scale data collection that scales easily and thus, in turns, enables the training of larger models.

- Thorough considerations for the data collection

A lot of thought has been put into the ways these data were collected, resulting in a dataset that contains sufficient information to learn from and guiding us in the ways how such data can be collected and scaled. In particular, the Authors have made the informed choices of the sensors (NO2, C2H5OH, VOC, CO, Alcohol, LPG), data collection modalities (temporal-difference), and standardization (controlling for the external air parameters; adding repetitions). All these choices were guided by direct data analyses.

- A comprehensive set of baseline models and metrics

The Authors have tested a comprehensive set of standard baseline models (MLP, CNN, RNN, Transformer) and metrics (top-1 and top-5 accuracy; F1-score) on three tasks (sensor-only; cross-modal; mixture) showing that (1) the labels are decodeable from the data and (2) that they are not saturated near 100%, leaving room for improvement. Both these properties are highly important for a benchmark dataset.

- Text clarity

The text is written clearly and structured nicely. The figures and the tables illustrate the concepts in the text well.

**Weaknesses:**

- The results are somewhat overstated

I found the results of the comparisons between the Transformer and the other models here to be overstated. While the Transformer surely shows some of the best results here, it's outperformed by the LSTMs and CNNs on several tasks (e.g. in Table 2). Likewise, I wouldn't say that "GC-MS supervision strongly boosts weaker models" -- I found the numbers that this statements refers to to be quite similar to each other, which is typical for transfer learning. Finally, I wouldn't focus on the claim that the dataset is much larger than the existing olfactory datasets, as these are two very different types of data: the latter involve psychophysics while the former one doesn't.

- Minor: The emphasis on the mixtures is unclear with this type of data

Typically, mixtures receive separate consideration in olfaction due to the differences in mass transfer between the molecules, differences in their affinity for the olfactory receptors, and synergetic effects in their perceptual qualities. None of these properties apply to the sensors or other the odor-emitting objects. Thus further discussion may be helpful to determine why the detection properties for the mixtures cannot be modeled as linearly dependent on the properties of individual smells.

**Questions:**

- What is the source of the temporal dynamics in the data?

While one of the results here regards the usefulness of the remporal data, it's unclear from the text where the temporal variations stem from. Is it because the sensor is brought closer to and then further from the odor sourse to mimic the inhalation? Or is it sue to the variability of the sensor readout at the constant distance from the odor source? In ether case, what do the temporal changes represent and why may they be important? What has motivated the collection of the temporal data in the first place?

- Could you please further comment on the utility of this type of data?

As this dataset abstracts form the brain and the processing witin it, the scope of the use case for this dataset differs from that of conventional olfactory datasets. Could you please detail the potential use cases for your type of data (some of them are mention in the introduction) and discuss the differences with the use cases for the conventional data?

- Minor: Why are C2H5OH and Alcohol different channels?

Seem to be the same thing to me.

---

> ### Author Response · Authors · 2025-11-21
> **Author's Response - Thank you for your feedback (part 1)**
>
> We thank the reviewer for the thoughtful and encouraging feedback. We are glad that the reviewer finds the paper well-written, clearly structured, and potentially impactful in an understudied area. Below we address the main concerns and clarifications.
>
> **On overstating some results and claims**
>
> We agree that our current wording can give the impression that ScentFormer dominates all other models, whereas in some settings LSTMs and CNNs perform similarly or slightly better. Our intention was to highlight that Transformer-style modeling is competitive and often strong on this dataset, not that it is uniformly superior. In the revision, we will soften these statements to reflect the actual numbers more accurately, like “Transformers achieve competitive or best performance across many configurations, but LSTMs and CNNs are close and sometimes slightly ahead.”
>
> We also agree that the phrase “strongly boosts” is too strong. For the MLP there are clear but modest gains. For CNN/LSTM/Transformer the changes are small and sometimes mixed, which is more typical of transfer supervision. We will revise this finding to something like “GC-MS supervision provides modest improvements for weaker models and only small, mixed effects for stronger temporal models,” and we now include additional experiments with a spectral GC-MS representation to show that this conclusion is robust to the specific encoding. This can be found in our other rebuttals.
>
> Finally, we agree that directly saying our dataset is “much larger” than conventional olfactory datasets can be misleading given that those involve psychophysical responses while ours does not. We will rephrase to emphasize that SmellNet is comparatively large for sensor based machine olfaction and explicitly distinguish it from psychophysical datasets, which are larger in terms of human-rated odorants but much smaller on the sensor/time-series side.
>
> **Mixtures and why they matter for this type of data**
>
> We appreciate the point that mixtures have a special status in olfactory neuroscience because of receptor binding, mass transfer, and perceptual interactions, and that some of those phenomena do not directly apply to our gas sensors.
> Our motivation for emphasizing mixtures is more application oriented. In real environments, odors rarely come from a single pure molecule or even a single ingredient. Kitchens, food storage, and consumer settings almost always involve overlapping sources. For SmellNet, we define mixtures at the ingredient level (e.g., apple + banana) with controlled mass ratios (25 to 75). Even if one assumes that the sensors’ responses are approximately linear in the partial pressures of volatiles, there are still nontrivial aspects to learn: the individual ingredients are themselves complex mixtures with overlapping volatile profiles; different sensors have different cross-sensitivities and saturation behaviors; environmental variables (temperature, humidity, airflow) modulate the effective sensor response.
>
> So while our mixtures do not probe receptor-level nonlinearities in the same way as mono-molecular mixtures in psychophysics, they do provide a realistic sensor-level mixture de-mixing problem: given a noisy superposition of responses from several complex sources, can we identify which ingredients and in what proportions are present? In the revision, we will make this distinction clearer and tone down any implication that we are directly addressing classical mixture questions in biological olfaction.
>
> **Source and motivation of temporal dynamics**
>
> We agree that our current description is too implicit. In our setup, the sensors are static and the temporal dynamics arise from a combination of the transient response of the sensors themselves (adsorption/desorption and thermal dynamics after exposure to a new odorant), mass transport of volatiles from the sample to the sensor (diffusion and local airflow as the sample is introduced and removed), and slower drift in environmental conditions (temperature, humidity, long-term sensor drift) even when the nominal setup is constant.
>
> Each recording typically includes a baseline period in clean air, followed by exposure to the odor source at a fixed position, during which the sensor responses rise and often partially saturate, and then a decay back toward baseline when the source is removed or the flow is changed. We collected temporal data because a single static snapshot (one reading per channel) would discard a lot of informative structure: rise times, slopes, and adaptation profiles differ across ingredients and mixtures, and in our experiments we indeed observe that models benefit from temporal windows and explicit temporal differencing.
> We will add a concise description of this temporal protocol in the methods section and clarify that the temporal dimension reflects both sensor kinetics and odor plume dynamics rather than deliberate movement of the sensor back and forth.

---

> ### Author Response · Authors · 2025-11-21
> **Author's Response - Thank you for your feedback (part 2)**
>
> Utility and use-cases of this type of data
> We agree that, because SmellNet abstracts away the brain and human reports, its scope differs from conventional psychophysical olfactory datasets, and we will make its sensor-side use cases more explicit. In particular, SmellNet supports robust source recognition in consumer and industrial settings (like identifying which ingredient or mixture is present in a production line, storage, or kitchen), mixture composition estimation for applications like quality control, counterfeit detection, and contamination monitoring, and systematic drift studies using the cross-day structure to evaluate how models cope with sensor and environmental variation over time. The large volume of time-series data also makes SmellNet a natural pretraining resource for perception-oriented models, where sensor encoders learned on SmellNet can later be aligned to human perceptual spaces using smaller psychophysical datasets. Most importantly for the longer-term vision, accurate mixture and source prediction from low-cost sensors is a prerequisite for smell recreation systems, in which an electronic nose would infer the composition of an ambient odor and drive an olfactory display to synthesize a matching or controllably modified scent for digital media, or accessibility applications. In the revision, we will explicitly contrast these sensor centric use cases with those of conventional psychophysical datasets and present SmellNet as a complementary proxy layer in the olfactory AI ecosystem.
>
> **C2H5OH vs. Alcohol channels**
> The reviewer is correct that chemically C2H5OH and alcohol refer to the same molecule ethanol. In our case, however, these correspond to two distinct commercially available sensors from the manufacturer, each with different nominal tuning and cross-sensitivity (one marketed specifically as an “alcohol” sensor and another labeled for ethanol with a different sensitivity curve). We retain both channels as separate inputs because, empirically, they exhibit slightly different response profiles across ingredients and mixtures. We agree that this can be confusing and will clarify in the hardware description that C2H5OH and Alcohol refer to two different physical sensor elements with overlapping but not identical selectivity, rather than two different chemical species.

---

> ### Comment · Reviewer_eaQT · 2025-11-22
> **Thank you**
>
> Thank you for your clarifications.
>
> My quesions have been addressed.
>
> Specifically:
>
> - The learning of the mixtures in the data is nontrivial because of various factors including different sensors having different cross-sensitivities and saturation behaviors
>
> - The temporal data is important because it's nonstationary, going from pure air to scent. In combination with the other point above (regarding the sensor saturation), temporal processing thus nicely seconds biological olfaction where it is important for the very same reason.
>
> I hope that the Authors would clarify both points in the text.
>
> With that, I keep my original high rating and the positive recommendation

---

> ### Author Response · Authors · 2025-12-03
>
> We thank the reviewer for the positive assessment and for summarizing the key points so clearly. We agree that:
> - Mixtures are nontrivial because of cross-sensitivity and saturation. Different sensors respond differently and nonlinearly to overlapping volatile profiles, so mixture decoding is more than just a linear combination of single-ingredient responses.
> - Temporal structure is essential because the signal is inherently nonstationary (baseline -> onset -> saturation/decay), and this makes temporal processing a natural analogue to biological olfaction, where dynamics during sniffing carry important information.
>
> In the camera-ready, we will explicitly highlight both points in the main text:
>
> - in the mixtures section, by emphasizing cross-sensitivity, saturation, and why this makes mixture learning a genuinely challenging task
> - in the methods/results sections, by clarifying the acquisition protocol and why non-stationary temporal dynamics are central to our models’ performance.
>
> We appreciate the reviewer’s high rating and recommendation.

---

### Official Review · Reviewer_uEQe · 2025-10-31

**Soundness:** 1
**Presentation:** 2
**Contribution:** 2
**Rating:** 2
**Confidence:** 4

**Summary:**

The authors propose SmellNet, a database of olfactory properties for both single-object and mixtures related to food. The authors gather this data for these objects placed in a controlled environment and use a metal oxide gas sensor to detect specific compounds, and track other ambient parameters alongside the data collection.  Using this data, the authors train a series of models for identifying the objects or mixture compositions via its sensory and ambient fingerprints. The authors evaluate the model’s performance to generalize across temporal domains for single objects, and also on unseen mixture compositions. Olfaction has long been a data-scarce domain, and I commend the authors’ efforts in bridging this gap for machine learning. However, from my perspective, there are some serious issues with the quality of the data, the methodology that the authors adopted, and the evaluation of the models proposed. Therefore, I am rejecting this paper, but am willing to raise my score if the authors address the numerous concerns I will detail below.

**Strengths:**

I think the writing is clear, and that the authors have posed their research questions in a structured format that makes it easy to understand the motivations for performing the experiments throughout the work.

The authors design a hardware platform for the collection of this data. Recognizing the limitations of sensor data, they attempt to pair sensory data up with high-quality GC-MS data for cross-modal learning. The authors also consider the role of sequential and non-sequential architectures for model performance.

**Weaknesses:**

I think the authors need to provide a stronger motivation for adopting gas sensors over other platforms beyond “portability”. Additionally, the authors justify the choices for the sensors as “common odors found in food, drinks and other common substances”. I am not convinced that typical food items contain large amounts of alcoholic, petrochemical and nitrogen oxide vapors, and it would be worrying if this was the case. For many of these natural products, the volatiles are mostly relatively complex organic compounds, which I am also not confident that the VOC sensor is able to capture beyond the fact that all these volatiles would be crammed into the same signal channel. Figure 9 in the SI also shows strong correlations for the NO2, C2H5OH and VOC sensors. Because the prior for choosing these sensors is not strong, I am worried that there is significant overfitting present.

Other evidence that points towards overfitting/data leakage exists in Table 16. Based on what I can see it seems possible to already build a reasonable classifier (e.g. tree-based model) just from temperature, pressure, humidity, gas resistance and altitude. Angelica for example can be classified by temperatures greater than >~26 and <33.59.

The GC-MS processing seems strange -- are the top 10 ingredients being completely broken up into its constituent atoms to form a representation? I had thought that the raw GC-MS data (i.e spectral data) was being used, but this form of considering only the atomic counts appears to be rather strange because the molecular structure will clearly affect its olfactory properties (and its ability to bind to the sensors). The PCA performed on top of the GCMS representation shows an extremely large PC1 importance -- and this could be because the ingredients are clearly separated by the presence/absence of certain elements, or atomic counts. Another slight weakness is that the GC-MS data from FooDB does not directly correspond to the same substance’s measurement under the same conditions (given natural variations in food products), so the gathered gas sensor data could potentially be incongruent with the GC-MS traces.

In terms of the utility of the models trained, querying requires measurement under the same pristine conditions and with the same sensor setup as well -- limiting its widespread use for classification. It would have been interesting to see this coupled to some data modality that is easily accessible to increase the scope of things that can possibly be predicted.

In alignment with the points above, I find it concerning that the authors claim that this dataset is relevant for the real-world when 1) the data was gathered in controlled environments to minimize environmental factors, which is not representative of the objects in the real-world, and 2) despite their efforts to control the environment they report wildly varying metrics for their environments.

I’m not entirely sure of the cost of model training, but testing the model could have been done across all other combinations of the leave-one-day-out approach the authors used for SmellNet-Base to show the robustness of their data modelling efforts.

The authors also report the limited generalizability of their approach in Table 4 for unseen combinations, but it’s not clear to me how these ratios were constructed, given that different masses of compounds lead to different concentrations and proportions of volatiles in the headspace, etc.

Finally, the authors evaluate the mixture proportion prediction task using a Top-x@0.1 metric. I’m not sure how 0.1 was decided, and if 0.1 is even an acceptable tolerance for error. The authors evaluate binary and ternary mixtures, and it’s not clear to me if weights are also predicted for more components than there are in the training set. The mixture ratios detailed in Appendix J are also rather limited and can almost be decomposed into a classification problem where the relevant odorant is identified within the mixture and its subsequent ratio-class is predicted. I think cosine similarity would be a more appropriate metric for this purpose.

**Questions:**

1) Do you have an explanation on why you chose to use gas sensors, and the specific sensors within your platform?
2) Please provide an ablation where either the sensory data or the environmental data is removed from the training data.
3) Why are the sensor outputs said to be qualitative (Section 4.2), but quantitative sensor readings are provided?
4) Please provide leave-day-out model performance metrics at least for SmellNet-Base.
5) What is the proportion of the GC-MS data that you could gather from FooDB? Is there an ablation where molecules that do not have the GC-MS data are eliminated, and the performance metrics are reported on a model trained with and without the GC-MS data?
6) Please show the performance in terms of cosine similarity for the mixture evaluation.
7) Is it surprising to you that the models can’t learn the time lag and that it is a requirement for you to inject this human-engineered feature into the model for increased model performance?

---

> ### Author Response · Authors · 2025-11-20
> **Author's Response - Thank you for your feedback (part 1)**
>
> We would like to thank the reviewer for their detailed comments and suggestions. We really appreciate it because we believe others would have similar concerns as well. We're more than happy to address the reviewer's concerns. Below are our responses to the questions:
>
> >1. Do you have an explanation on why you chose to use gas sensors, and the specific sensors within your platform?
>
> We thank the reviewer for raising this question. Beyond portability, an important reason we chose metal-oxide (MOX) gas sensors is that they are inexpensive and easy to integrate into embedded systems. One of our primary goals is to build a platform that can eventually be deployed broadly, which would be expensive with GC-MS. Based on current market prices, our entire system can be built from off-the-shelf components for roughly \$100 to \$150 per unit, making large-scale, distributed deployments feasible [1-4].
>
> Another reason is that the MOX sensors have broad and overlapping cross-sensitivity to many VOCs. In many applications, cross sensitivity is viewed as noise, but in our case it is what makes each channel informative for complex odors. Each MOX sensor responds to a broad family of volatiles, so each channel provides a noisy, overlapping projection of the underlying volatile profile. The combination of multiple such sensors provides enough structure for a learning model to distinguish different substances, as we demonstrate in our classification experiments. This is an analogy to biological olfactory receptors, which are broadly tuned rather than one molecule per receptor.
>
> Regarding the specific channels, these are just names for manufacturer calibration labels, not an assumption that foods emit large quantities of NO2 or pure alcohol. In practice, each MOX sensor responds to a wide range of oxygenated and hydrocarbon volatiles. We chose this set because together they span common chemical families found in food and are standard in low-cost e-nose designs [5-7]. The strong correlations the reviewer notes between NO2, C2H5OH and VOC channel reflect the fact that many foods co-emit these volatile classes under our controlled conditions, not that the sensors are measuring entirely unrelated phenomena.
>
> Finally, we include separate environmental sensors so that models can explicitly account for ambient conditions rather soaking them into the gas channels. In the next section, we analyzed both chemical and environmental channels, and we will make this motivation clearer in the revised text.
>
> > Please provide an ablation where either the sensory data or the environmental data is removed from the training data.
>
> We thank the reviewer for requesting clarification on sensory and environmental data. We ran an ablation where we separately train on gas sensor channels only, environmental channels only (temperature, pressure, humidity, and altitude), and all channels. Table 1 reports results for ScentFormer on SmellNet-Base (window size = 100, gap = 0, 25).
>
> **Table 1:** Ablation of ScentFormer on SmellNet-Base (w = 100 and g = 0, 25) using sensory data only, environment data only, and all data. Sensory data clearly dominate performance, and environment only models are much weaker.
> | Input type | g  | Acc@1 | Acc@5 | F1    |
> |-----------------------|----|-------|-------|-------|
> | Sensory only | 0  | 38.0  | 79.1  | 35.0  |
> | Sensory only | 25 | 62.5  | 91.2  | 60.7  |
> | Env only | 0  | 21.5  | 57.1  | 18.3  |
> | Env only | 25 | 21.9  | 65.9  | 19.4  |
> | Total (all channels) | 0  | 45.0  | 81.7  | 40.6  |
> | Total (all channels) | 25 | **71.3** | **96.8** | **70.6** |
>
> Compared to the full model, an environmental data only model drops from 45-71.3% Acc@1 to 21.5-21.9%, which a sensor only model retains most of the performance. This shows that environmental variables alone are not enough to identify the substance, and the main signal comes from the gas sensors. Environment features act as a modest supplement rather than a primary signal or shortcut.

---

> ### Author Response · Authors · 2025-11-20
> **Author's Response - Thank you for your feedback (part 2)**
>
> **(continued author's response)**
>
> > Please provide an ablation where either the sensory data or the environmental data is removed from the training data.
>
> **Table 2:** Ablation of a tree-based classifier (Random Forest) on SmellNet-Base using environment data, sensor data, and all channels for different window sizes w and temporal gaps g. All numbers are percentages. Environmental only models are consistently much weaker than models with gas sensors, confirming that our conclusions do not depend on a particular architecture.
>
> | w   | g   | Input type  | Acc@1    | Acc@5    | F1 (macro) |
> |-----|-----|-------------|----------|----------|------------|
> | 50  | 0   | Env only    | 15.80    | 46.25    | 13.05      |
> | 50  | 0   | Sensor only | 37.07    | 73.26    | 35.95      |
> | 50  | 0   | Full (all)  | **43.95** | **80.23** | **40.98** |
> | 50  | 25  | Env only    | 10.90    | 39.70    | 8.90       |
> | 50  | 25  | Sensor only | 42.38    | 80.24    | 41.56      |
> | 50  | 25  | Full (all)  | **48.48** | **86.15** | **47.51** |
> | 100 | 0   | Env only    | 16.57    | 47.08    | 13.77      |
> | 100 | 0   | Sensor only | 36.16    | 72.88    | 34.61      |
> | 100 | 0   | Full (all)  | **44.07** | **80.23** | **41.56** |
> | 100 | 25  | Env only    | 9.76     | 39.84    | 7.57       |
> | 100 | 25  | Sensor only | 42.63    | 80.48    | 40.87      |
> | 100 | 25  | Full (all)  | **46.61** | **86.85** | **45.68** |
>
> Across all (w, g) settings and for this non-neural model, environmental only (9.76%) performance is far below sensor only (42.63%) and full channels (46.61%). Thus, there is no evidence that environmental variables provide an easy shortcut to the labels. Models need the gas sensor channels to perform well, and environment features add only a modest incremental benefit.
>
> We will add this ablation study along with the table in the appendix and clarify in the main text that performance is driven by the smell sensors, and environmental variables only provide a weak signal to the model as a supplement.
>
> > Why are the sensor outputs said to be qualitative (Section 4.2), but quantitative sensor readings are provided?
>
> We thank the reviewer for pointing out this ambiguity and apologize for the confusing wording. By "qualitative rather than absolute quantitative" in Sec 4.2, we did not mean that the sensors output discrete categories, and the raw readings are real value time series. What we intended to convey is that our MOX sensors are uncalibrated. Their outputs depend on sensor-to-sensor variation, non-linear response curves, cross-sensitivity, and ambient conditions. As a result, a value of 200 on a given channel cannot be interpreted as 200 ppm of a specific compound, nor is it directly comparable across devices without additional calibration.
>
> In this sense, the sensor outputs are qualitative or "semi-quantitative" values of the underlying volatile profile. They reliably encode relative patterns (e.g., “ingredient A consistently elicits a stronger response than ingredient B on this channel”, or “the signal increases and then decays over time”), but they are not physically meaningful absolute measurements in the way that GC-MS concentrations are. This is why, in Sec 4.2, we emphasize temporal differences and standardization. As a result we can see, when we apply a temporal difference on the data, the performance get significantly better for most models.
>
> We will update the text to make this distinction clearer by rephrasing “qualitative rather than absolute quantitative” as “semi-quantitative, uncalibrated readings that are reliable in a relative sense but not calibrated absolute concentrations,” and by explicitly contrasting them with the calibrated GC-MS measurements.

---

> ### Author Response · Authors · 2025-11-21
> **Author's Response - Thank you for your feedback (part 3)**
>
> > Please provide leave-day-out model performance metrics at least for SmellNet-Base.
>
> We thank the reviewer for raising the concern about temporal robustness. Thus, we ran extra experiments for leave-one-day-out performance on SmellNet-Base.
>
> **Table 3** Leave-one-day-out performance of ScentFormer on SmellNet-Base
> for g = 25. We report mean ± standard deviation over the
> 6 held-out days for each window size. All numbers are percentages.
>
> | w| Acc@1 (%)| Acc@5 (%)| F1 (macro, %)|
> |-----|-----------------|-----------------|------------------|
> | 50| 49.4 ± 8.0| 84.2 ± 6.3| 48.2 ± 7.9|
> | 100| **53.0 ± 6.0** | **85.5 ± 5.0**| **51.6 ± 6.0**|
>
> **Table 4.** Per-day leave-one-day-out performance of ScentFormer on SmellNet-Base
> for g = 25 and w = 50, 100. Each row corresponds to holding out one day as the test set. All numbers are percentages.
>
> | w| Day | Acc@1 | Acc@5 | F1 (macro) |
> |-----|-----|-------|-------|-----------|
> |50| 1   | 34.8  | 73.5  | 33.3|
> |50 | 2   | 54.7  | 90.7  | 52.8|
> |50  | 3   | 55.7  | 89.4  | 54.2|
> |50  | 4   | 54.5  | 85.9  | 53.2|
> |50  | 5   | 45.9  | 80.7  | 46.3|
> |50  | 6   | 50.6  | 85.0  | 49.5|
> |100 | 1   | 43.6  | 80.8  | 41.4|
> |100 | 2   | 56.0  | 89.6  | 54.9|
> |100 | 3   | 57.1  | 89.5  | 54.1|
> |100 | 4   | 58.0  | 88.0  | 56.6|
> |100 | 5   | 47.2  | 77.9  | 47.0|
> |100 | 6   | 56.1  | 87.4  | 55.5|
>
> In our setup, each day in SmellNet-Base corresponds to a separate acquisition session on a different calendar day with different ambient conditions. In the leave-one-day-out protocol, we hold out all samples from one day as the test set and train on the remaining five, so the model must generalize across day-to-day shifts rather than relying on iid resampling.
>
> We note that Day 1 is consistently the most challenging day. For all (w, g) configurations, Day 1 gives the lowest Acc@1 and macro-F1. We suspect this is due to ambient and sensor state differences on that day, which create a larger distribution shift between Day 1 and the remaining days. Importantly, even on this hardest day the model remains well above chance (2% Acc@1 for 50 classes), and on the other five days performance clusters in the mid-50% range for Acc@1 under our best configuration. Overall, the mean $\pm$ std over days reflects genuine between-day domain differences rather than instability of the training procedure, and shows that ScentFormer generalizes robustly across realistic day-to-day variations in SmellNet-Base.
>
> We will add these leave-one-day-out results to the paper Appendix and briefly summarize them in the main text. We will also release our dataset with labeled dates.
>
> > What is the proportion of the GC-MS data that you could gather from FooDB? Is there an ablation where molecules that do not have the GC-MS data are eliminated, and the performance metrics are reported on a model trained with and without the GC-MS data?
>
> **GC-MS data source and coverage**
> We appreciate the reviewer’s question. In our current experiments, all ingredients used in SmellNet were chosen specifically because they appear in FooDB with annotated volatile compounds and GC-MS information. In other words, the coverage of GC-MS descriptors for the 50 base substances is 100%. We do not performa any GC-MS measurements ourselves. Instead, we treat FooDB as an ingredient level prior over typical volatile composition.
>
> Because all of these ingredients have GC-MS descriptors, the requested ablation "removing molecules without GC-MS" would leave the label set unchanged. Instead, our ablation is between sensor only training and sensor + GC-MS contrastive supervision on exactly the same 50 ingredients. In the original paper, the “sensor only” columns (RQ1) correspond to training without GC-MS, and the “cross-modal” columns (RQ2) in Table 1 correspond to training with GC-MS on the same classes. In the revision we will make this explicit.
>
> **Revised GC-MS representation (raw spectra instead of atomic counts)**
> We agree that the original element count representation was unusual. In the revision, we replace it with a standard spectral fingerprint. For each ingredient, we, collect its volatile compounds from FooDB, retrieve all available experimental Electron Ionization mass spectra for those compounds, and convert each spectrum into a vector of binned intensities over 40-500 m/z (1 Da bins, max-normalized). We then average spectra per compound and across compounds to obtain a fixed length GC-MS embedding for each ingredient. The sensor encoder, training schedule, and evaluation protocol remain unchanged. Only the GC-MS side is replaced by this spectral representation. We decided on the 40-500 m/z because this range covers 99% of the intensities in our scraped data.

---

> ### Author Response · Authors · 2025-11-21
> **Author's Response - Thank you for your feedback (part 4)**
>
> (continued author's response)
>
> > What is the proportion of the GC-MS data that you could gather from FooDB? Is there an ablation where molecules that do not have the GC-MS data are eliminated, and the performance metrics are reported on a model trained with and without the GC-MS data?
>
> **Table 5.** Comparison of single-ingredient classification on SmellNet-Base with three GC-MS settings: sensor-only baseline (no GC-MS, labeled S), cross-modal training with element count GC-MS embeddings ($X_{atom}$), and cross-modal training with spectral GC-MS embeddings ($X_{spec}$). We report Top-1 accuracy (Acc@1, %) for each model, window size w, and temporal gap g. $Δ_{atom}$ and $Δ_{spec}$ denote the change in Acc@1 relative to the sensor-only baseline. Spectral GC-MS supervision shows the same qualitative behavior as the original element-count encoding, where largest gains for MLPs and small, mixed effects for stronger temporal models.
>
> | Model       | w   | g   | S | $X_{atom}$ | $X_{spec}^*$ | $Δ_{atom}$ | $Δ_{spec}^*$ |
> |------------|-----|-----|-----------------|--------|--------|--------|--------|
> | MLP        | 50  | 0   | 21.9| 23.6   | 24.1   | +1.7   | +2.2   |
> | MLP        | 50  | 25  | 18.2| 23.8   | 24.7   | +5.6   | +6.5   |
> | MLP        | 100 | 0   | 21.0| 25.6   | 24.3   | +4.6   | +3.3   |
> | MLP        | 100 | 25  | 26.8| 28.0   | 28.5   | +1.2   | +1.7   |
> | CNN        | 50  | 0   | 25.5| 28.4   | 27.7   | +2.9   | +2.2   |
> | CNN        | 50  | 25  | 46.9| 45.9   | 45.7   | −1.0   | −1.2   |
> | CNN        | 100 | 0   | 29.5| 31.0   | 26.7   | +1.5   | −2.8   |
> | CNN        | 100 | 25  | 52.7| 57.1   | 55.0   | +4.4   | +2.3   |
> | LSTM       | 50  | 0   | 29.3| 29.7   | 31.9   | +0.4   | +2.6   |
> | LSTM       | 50  | 25  | 50.6| 53.3   | 50.7   | +2.7   | +0.1   |
> | LSTM       | 100 | 0   | 28.8| 33.7   | 35.2   | +4.9   | +6.4   |
> | LSTM       | 100 | 25  | 57.9| 56.1   | 56.2   | −1.8   | −1.7   |
> | Transformer| 50  | 0   | 35.1| 36.2   | 34.8   | +1.1   | −0.3   |
> | Transformer| 50  | 25  | 50.6| 50.9   | 52.6   | +0.3   | +2.0   |
> | Transformer| 100 | 0   | 39.9| 41.0   | 39.5   | +1.1   | −0.4   |
> | Transformer| 100 | 25  | 56.1| 58.5   | 55.8   | +2.4   | −0.3   |
>
> \* means new to this rebuttal
>
> Across all architectures, the GC-MS spectral supervision shows the same qualitative behavior as our original GC-MS element count method. The weaker MLP benefits the most, while CNN/LSTM/ScentFormer see only small, sometimes mixed improvements. Our main RQ2 conclusions are still robust to the choice of GC-MS encoding.
>
> Taken together, these results show that our RQ2 conclusions are robust to the choice of GC-MS encoding. Whether we use an element count vector or a full spectral representation, GC-MS functions as an optional chemistry aware prior that modestly regularizes the sensor embedding, and the main performance gains still come from modeling the sensor time series.
>
> **How GC-MS is used in the model**
> GC-MS data enter the model only through ingredient-level embeddings. For each ingredient, we precompute a GC-MS embedding from FooDB. During training, we use a CLIP style contrastive loss. Sensor windows for an ingredient are pulled closer to that ingredient’s GC-MS embedding and pushed away from others. At inference, we do not require any new GC-MS measurement. We embed the sensor window and classify it by comparing it to the fixed set of GC-MS label embeddings.Thus, GC-MS acts as a label representation and chemistry aware prior rather than as an additional per-sample input modality.
>
> **Why also consider the original element count representation**
> In our original submission we converted compound formulas to element counts to obtain a simple descriptor that is insensitive to retention times and chromatographic conditions. This representation captures coarse compositional trend and was used only as a supervisory signal, not as the final feature space. As the reviewer notes, however, it inevitably discards fine-grained spectral structure and can lead to a dominant first principal component. In the revision, we added an alternative GC-MS representation based on raw-spectra binning (40–500 m/z, 1 Da bins) and show that both encodings lead to consistent qualitative conclusion. GC-MS supervision is a complementary signal that modestly regularizes the sensor embedding, but the bulk of performance comes from modeling the sensor time series itself. We will also include example spectra for several ingredients in the appendix for transparency.

---

> ### Author Response · Authors · 2025-11-21
> **Author's Response - Thank you for your feedback (part 5)**
>
> > Please show the performance in terms of cosine similarity for the mixture evaluation.
>
> **Table 6** Mixture proportion prediction performance on `SmellNet-Mixture` for seen and unseen mixtures. We report KL divergence, mean absolute error (MAE), and cosine similarity between the predicted and ground truth 12-D mixture distributions for different models and window sizes. Cosine similarity stays high for both seen and unseen mixtures, while KL and MAE clearly reveal the degradation in ratio accuracy on unseen combinations.
>
> | Mixture | w| Model| KL| MAE| Cosine |
> |---------|-----|-------------|-------|--------|--------|
> | seen| 50| MLP| 0.450 | 0.0428 | 0.861  |
> | seen| 50| CNN| 0.370 | 0.0404 | 0.867  |
> | seen| 50| LSTM| 0.416 | 0.0399 | 0.878  |
> | seen| 50| ScentFormer | 0.480 | 0.0395 | 0.863  |
> | seen| 100| MLP| 0.615 | 0.0586 | 0.799  |
> | seen| 100| CNN| 0.425 | 0.0476 | 0.849  |
> | seen| 100| LSTM| 0.503 | 0.0430 | 0.856  |
> | seen| 100| ScentFormer | 0.386 | 0.0417 | 0.874  |
> | unseen  | 50  | MLP| 2.919 | 0.1190 | 0.860  |
> | unseen  | 50  | CNN| 2.649 | 0.1160 | 0.860  |
> | unseen  | 50  | LSTM| 2.532 | 0.1160 | 0.848  |
> | unseen  | 50  | ScentFormer | 2.618 | 0.1100 | 0.863  |
> | unseen  | 100 | MLP| 2.302 | 0.1110 | 0.799  |
> | unseen  | 100 | CNN| 2.488 | 0.1160 | 0.824  |
> | unseen  | 100 | LSTM| 2.441 | 0.1150 | 0.856  |
> | unseen  | 100 | ScentFormer | 2.454 | 0.1110 | 0.895  |
>
> **Mixture ratios and task definition**
> Each Smell-Mixture sample is defined by a volumetric mixing ratio over 12 base odorants (e.g. 50/50 binaries; 10/30/60 ternaries). We normalize these recipes to a 12-D vector and train the model to predict this vector from the sensor time series. The goal is to recover the intended recipe proportions, not exact gas-phase concentrations, which depend on volatility and headspace physics.
>
> **Why Top-1@0.1 (and not only cosine)**
> Ratios in the dataset are defined on a 0.1 grid, so a ±0.1 tolerance corresponds to being within one step of the underlying recipe resolution. A smaller threshold would be below the precision of our mixing protocol, while a much larger one would not distinguish, for example, 30:70 from 50:50. For this reason we use Top-1@0.1 (and dynamic Top-K) together with MAE, and now additionally report cosine similarity as requested.
>
> **Cosine similarity results**
> As Table 6 shows, cosine similarity is high for seen mixtures and remains high for unseen mixtures. However, KL and MAE increase substantially and Top-1@0.1 drops sharply on unseen ratio, confirming our original conclusion that precise ratio extrapolation to novel combinations is hard even when the model roughly identifies the right components. Cosine is a useful sanity check for overall directional agreement between predicted and true mixture vectors, but less discriminative than MAE and Top-1@0.1 quantifying the absolute errors in mixture proportions.
>
> **Mixture prediction vs. classification**
> Although we only evaluate on mixtures with 1 to 3 non-zero components, the model always predicts a continuous 12-D distribution and is trained with KL/MAE/other loss functions, not a small set of discrete ratio classes. The binary and ternary ratio cover a range of grids and asymmetries (e.g. 10:90, 10:30:60, 33:33:33), so the task requires learning continuous proportions rather than just which odorant. Extending to mixtures with more than three components is a natural next step once such data is collected.
>
> We will incorporate Table 6 and this analysis into the revised mixture section, and add the full cosine/MAE/KL metrics for all models and settings to the appendix in the camera-ready version.
>
> > Is it surprising to you that the models can’t learn the time lag and that it is a requirement for you to inject this human-engineered feature into the model for increased model performance?
>
> Actually, we do not find this result surprising. In principle, a temporal model could approximate a lagged difference internally, but in practice, it is standard in time-series work to use simple, domain motivated transforms (e.g. delta features in speech recognition [8, 9], log-returns in finance [10, 11]) before learning. For MOX sensors in particular, relative changes are more informative than absolute levels, which are affected by drift and device-specific offsets, and slow baseline shifts. Our results at g = 0 already show that sequential architectures can exploit raw temporal structure (they clearly outperform MLP), but adding an explicit lag makes these informative differences easier to access and improves optimization and sample efficiency. We therefore view the lag as a generic, physically motivated preprocessing step rather than a task-specific crutch, and note that it does not change the relative ranking or qualitative behavior of the models, only the overall performance level. We will clarify this motivation in Sec. 4.2 and explicitly discuss lagged differences as a standard preprocessing choice in time-series sensing.

---

> ### Author Response · Authors · 2025-11-22
> **References**
>
> [1] Adafruit Industries. “MQ-3 Alcohol Gas Sensor Module.” Product page, accessed November 2025. Low-cost metal-oxide gas sensor module for alcohol/solvent vapors, with typical single-unit retail price in the low–single-digit US\$ range.
>
> [2] Adafruit Industries. "MQ-135 Air Quality / VOC Gas Sensor Module." Product page, accessed November 2025. Metal-oxide sensor module targeting air quality and VOCs, commonly sold for a few US\$ per unit.
>
> [3] Adafruit Industries. “BME280 or SPI Temperature, Humidity, and Barometric Pressure Sensor Breakout.” Product page, accessed November 2025. Combined environmental sensor breakout board, typically priced in the US\$5–10 range.
>
> [4] Espressif Systems. “ESP32-DevKitC Development Board.” Product page / distributor listing, accessed November 2025. Low-cost Wi-Fi/Bluetooth microcontroller board, typically priced in the US\$5–15 range.
>
> [5] Rodríguez, Juan, et al. “Electronic nose for quality control of Colombian coffee through the detection of defects in ‘Cup Tests’.” Sensors 10.1 (2010): 36–46.
>
> [6] Wijaya, Dedy Rahman, Riyanarto Sarno, and Enny Zulaika. “Electronic nose dataset for beef quality monitoring in uncontrolled ambient conditions.” Data in Brief 21 (2018): 2414–2420.
>
> [7] Mahradian, Mehrab. “Fruit Freshness, Electronic Nose Dataset.” Kaggle dataset (2023).
>
> [8] Furui, Sadaoki. "Speaker-independent isolated word recognition using dynamic features of speech spectrum." IEEE Transactions on Acoustics, Speech, and Signal Processing 34.1 (1986): 52–59.
>
> [9] Young, Steve, et al. "The HTK Book (for HTK Version 3.4)." Cambridge University Engineering Department (2006).
>
> [10] Tsay, Ruey S. Analysis of Financial Time Series. 3rd ed., Wiley, 2010.
>
> [11] Campbell, John Y., Andrew W. Lo, and A. Craig MacKinlay. The Econometrics of Financial Markets. Princeton University Press, 1997.

---

> > ### Comment · Reviewer_uEQe · 2025-11-23
> >
> > Thanks to the authors for providing an extremely detailed rebuttal -- I will raise my score to 6. Many of these rebuttal results are rather surprising (e.g. how poorly Env Only performed as a classifier when it looked like it could have performed better, spectral fingerprints performing equivalently to bag-of-atoms approach for ). The authors have satisfied many of my concerns, though I will still maintain that the experimental protocol for gathering the data needs to be much more robust where some kind of environmental subtraction is considered. At this point I believe that the same material gathered in a different environment might not receive the same performance.
> >
> > I also get the authors' point about the cross-sensitivity, but given the extremely large correlation between the three aforementioned sensors, I think its possible to drop any one of them and still achieve the same predictive performance -- i.e the inclusion of all three channels does not add significant information even if we know that a volatile triggers all three sensors.

---

> > > ### Author Response · Authors · 2025-12-03
> > >
> > > We thank the reviewer again for their detailed reading of our rebuttal and for raising their score.
> > >
> > > We agree that robustness to environmental variation is a key limitation of our current dataset and protocol. In the camera-ready, we will make this explicit in the discussion and position SmellNet more like an early benchmark (e.g., MNIST, CIFAR-style) for sensor-based olfaction in a controlled but realistic setting, rather than a fully solved solution for arbitrary environments. We also plan to highlight explicit background and environment subtraction and collection in more diverse settings as primary directions for follow-up data collection.
> > >
> > > On cross-sensitivity and the three highly correlated channels, we agree that some redundancy is expected. Our ablations (environment-only vs. sensor-only vs. all channels, and single-channel masking in the Appendix H.2 Tables 10–11) already show that the gas-sensor array as a whole provides the dominant signal, while no single channel acts as a shortcut. In the camera-ready, we will surface these results more clearly and temper our claims. We do not argue that every individual sensor is indispensable, but that a small, slightly redundant MOX array provides a practical and robust basis for low-cost machine olfaction.
> > > We appreciate the reviewer’s suggestions and will incorporate these clarifications and limitations into the final version.

---

> ### Author Response · Authors · 2025-12-03
> **Thanks for raising the score**
>
> We sincerely thank reviewer uEQe for actively engaging in our rebuttal and raising the score to 6. We will incorporate all feedbacks in our final manuscript.

---

### Official Review · Reviewer_WutW · 2025-11-01

**Soundness:** 2
**Presentation:** 3
**Contribution:** 2
**Rating:** 6
**Confidence:** 2

**Summary:**

This paper presents SMELLNET, a dataset of gas sensor recordings collected from 50 natural substances along with mixtures among them. The dataset contains approximately 828,000 time-series data points and is paired with preexisting GC-MS data. The authors also introduce SCENTFORMER, a Transformer-based model designed to process these temporal signals for substance classification and mixture ratio prediction. The authors claim that SMELLNET can serve as a benchmark for research in olfactory AI and AI-based smell recognition

**Strengths:**

- The paper is well-written and clearly structured, making it easy to follow.
It explores an understudied and potentially impactful area, bridging olfactory perception and machine learning.
- The introduction of a new dataset and benchmark is a valuable contribution, especially in a domain with limited data resources.

**Weaknesses:**

- The dataset includes no human perceptual data (e.g., semantic descriptors, intensity or pleasantness ratings, similarity judgments, or brain recording). Without human responses, the dataset cannot be meaningfully linked to olfactory AI studies as claimed in the paper. Labeling the dataset as “SMELLNET” and positioning it as central to “olfactory perception” is overstated. At best, the dataset captures chemical sensing information, not perceptual smell data.
- In Table 1, the comparison with human-evaluated datasets (e.g., Dravnieks, DREAM, Snitz, Ravia) is misleading. Other datasets include both stimulus (odorants) and response (human judgments). SMELLNET only includes the stimulus component (gas sensor signals), not perception-based responses. Comparing against other gas-sensor or e-nose datasets that exist in related areas would probably be more relevant and better contextualize the scale and novelty of SMELLNET.
- Although the paper describes the dataset as “large-scale,” 50 odorants with six repetitions each is modest by modern machine learning standards. Summing the total number of time-series data points to claim large-scale is not meaningful. A more relevant metric would include the number of unique odorants and repetitions.
- The definition of “mixture” is ambiguous and the motivation behind that is not clear. In olfactory research, mixtures typically refer to combinations of mono-molecular odorants, while here it seems to involve mixing natural extracts (e.g., “apple + banana”), which inherently are mixtures themselves. Predicting this kind of mixture composition from sensor data, without any perceptual correspondence, does not provide meaningful insight into olfactory AI.
- The dataset provides low-resolution sensor data, yet it later depends on pairing with high-resolution GC–MS data. It is unclear why the low-resolution data is necessary or what advantages it offers. No ablation is provided to show how a model trained solely on high-resolution data would perform compared to one using sensor data.

**Questions:**

- What concrete research questions in olfactory AI can this dataset help answer?
- How is the mixture dataset can be used in the further research
-How are “mixtures” defined chemically?
- Could the authors provide comparisons between SMELLNET and other gas-sensor or GC–MS datasets?
- How do you envision connecting SMELLNET to actual human olfactory perception in future work?

---

> ### Author Response · Authors · 2025-11-21
> **Author's Response - Thank you for your feedback (part 1)**
>
> We thank the reviewer for the thoughtful and encouraging feedback. We are glad that the reviewer finds the paper well-written, clearly structured, and potentially impactful in an understudied area. Below we address the main concerns and clarifications.
>
> **Scope: olfactory AI vs chemical sensing and naming of SmellNet**
>
> We agree that our current wording can blur the distinction between human olfaction perception and sensor based machine olfaction. SmellNet, as it stands, contains only stimulus-side information. It does not include human semantic ratings, intensity, pleasantness, similarity judgments, or neural data. In the revision we will explicitly present SmellNet as a dataset for sensor based smell recognition or machine olfaction, rather than as a direct dataset of human perceptual responses. We will soften language in the title, abstract, and introduction that suggests centrality to “olfactory perception,” and instead position SmellNet as a hardware-proximal resource that is complementary to human-rated olfactory datasets. In the discussion section, we will also outline how SmellNet can be used as a building block for future work that links sensor representations to human perception (see our response to the “future work” question).
>
> **Table 1: comparison to human-evaluated datasets (Could the authors provide comparisons between SmellNet and other gas-sensor or GC-MS datasets?)**
>
> We agree that placing SmellNet in the same block as perception datasets such as Dravnieks, DREAM, Snitz, or Ravia can be misleading, since those resources contain both stimuli and human responses, whereas SmellNet currently contains only the stimulus side (gas-sensor time series plus GC-MS derived descriptors).
>
> In the revision we will clearly separate human-perception datasets (odorant + human response) from sensor datasets (odorant + sensor) in Table 1. We will also make explicit in the caption and text that SmellNet is a sensor resource, not a psychophysical one. The comparison to perception datasets will be framed as showing complementary modalities and relative scale, rather than suggesting equivalence.
>
> In addition to human‐evaluated olfactory datasets, there is a small but growing body of work on gas-sensor and e-nose datasets. Examples include e-noise recordings for Colombian coffee quality control [1], beef quality monitoring in uncontrolled ambient conditions [2], a fruit freshness dataset collected with MQ series MOS sensors [3], and generic gas sensor array benchmarks that focus on drift and calibration, such as the Chemical Gas Sensor Array Dataset and the Gas Sensor Array Drift Dataset at Different Concentrations [4,5]. These datasets are valuable precedents, but they typically target a single product category, do not include controlled multi-ingredient mixtures, and are not paired with GC-MS derived descriptors. SmellNet is designed to complement this literature by providing commodity-sensor time series for 50 food ingredients and 43 mixtures, recorded across multiple days and aligned with ingredient level GC-MS embeddings.
>
> We will emphasize this positioning in the revised table and related work section.
>
> **"Large-scale" claim and dataset size**
>
> We appreciate the concern about calling 50 odorants with six repetitions large-scale in an absolute ML sense. Our intent was “large” relative to existing sensor based datasets, most of which involve a single product type with tens of samples and shorter recordings.
>
> In the camera-ready we will avoid unqualified claims of "large-scale", rephrase our wording to "comparatively large for sensor-based machine olfaction", and spell out what this means concretely: 50 natural substances, 43 controlled multi-ingredient mixtures, 68 hours of recordings and 828k time steps, collected across multiple days to enable drift and robustness studies. We will also update Table 1 to emphasize more meaningful scale metrics (number of unique odorants, mixtures, and repetitions) and explicitly position SmellNet alongside existing coffee, beef, fruit-freshness, and UCI gas-sensor datasets, so that its size is clearly contextualized rather than presented as large in an absolute sense.

---

> ### Author Response · Authors · 2025-11-21
> **Author's Response - Thank you for your feedback (part 2)**
>
> **Mixture definition and motivation (How is the mixture dataset can be used in the further research - How are “mixtures” defined chemically?) **
>
> We agree that our notion of "mixtures” is different from classic olfactory studies that mix mono-molecular odorants. In SmellNet, mixtures are defined as combinations of natural ingredients (e.g., apple + banana), each of which is itself a complex mixture of volatile compounds. We will clarify that our primary application focus is real-world ingredient mixtures, where odor objects (meals, beverages, product formulations) are almost always composite.
>
> Concretely, each mixture in SmellNet is defined by a mass ratio over the contributing ingredients (e.g., 25:75, 10:30:60). In the current paper, all mixture experiments use only the gas-sensor time series and the ingredient mass ratios as supervision. GC-MS is not used for training or evaluating the mixture models. In future work, for ingredients with FooDB entries, one could approximate a mixture’s volatile composition as a weighted combination of their GC-MS embeddings and study how well sensor-based mixture representations align with such chemical priors.
>
> In the revision, we will avoid suggesting that we directly address receptor-level questions about mono-molecular mixtures. Instead, we will position the SmellNet-Mixture benchmark as targeting practical machine-olfaction questions, mixture detection and composition recovery for everyday foods with low-cost sensors, while pointing to GC-MS based mixture modeling as a natural extension for future work.
>
> **Low-resolution sensors vs. high-resolution GC-MS**
>
> Our main goal is to enable portable, low-cost smell recognition with commodity gas sensors. High-resolution GC-MS is accurate but expensive, slow, and not suitable for embedded or always-on scenarios. This is why we focus on the low-resolution, noisy sensor data as the primary modality.
>
> GC-MS plays a different role. It serves as an ingredient-level chemical prior: for each ingredient, we precompute a single GC-MS embedding from FooDB. We use this embedding only for label-side supervision (contrastive training between sensor embeddings and GC-MS embeddings). It is never required as a per-sample input at test time.
>
> To clarify the relationship further, in the revision we will explicitly state that our core task is sensor-based recognition, and GC-MS is only used to structure the embedding space. Our main contribution is the lower resolution sensor data, which is available to collect anytime. We only use the high resolution GC-MS as a supplement to the higher quality data.
>
> **What concrete research questions in olfactory AI can this dataset help answer?**
>
> SmellNet is designed to support several concrete research questions in olfactory AI, specifically on the sensor side. First, it enables studying how well models can generalize across days and environmental drift when trained on gas-sensor time series, via leave-day-out splits. Second, by pairing each ingredient with a GC-MS–derived embedding, it allows investigation of how effectively low-cost sensors can approximate high-resolution chemical structure through contrastive, chemistry-aware training. Third, the mixture subset (50 ingredients, 43 mixtures with controlled mass ratios) allows testing whether models can recover ingredient identities and proportions from real-world food mixtures rather than only from single, pure substances. Accurate mixture prediction is a key prerequisite for future smell recreation systems, where an electronic nose would infer a composition from the environment and drive an olfactory display to synthesize a matching scent, enabling applications for downstream smell recreation devices. Finally, because we include multiple architectures and preprocessing variants, SmellNet provides a benchmark for comparing temporal modeling strategies (MLP vs. CNN vs. LSTM vs. ScentFormer) and for analyzing which sensor channels and environmental variables are most important for robust smell recognition.
>
> **How do you envision connecting SmellNet to actual human olfactory perception in future work?**
>
> As a concrete next step toward perception, we plan to annotate each ingredient in SmellNet with a small set of perceptual attributes (e.g. sweet, sour, fruity, smoky, sulfurous) based on existing psychophysical descriptions and expert curation. Using these annotations, we can treat ScentFormer as a frozen encoder and train simple linear probes to predict the presence or absence of each attribute from the sensor embeddings. If these perceptual tags can be decoded reliably above chance, this would indicate that the learned sensor representations capture structure that is at least partially aligned with human perceptual dimensions.

---

> ### Author Response · Authors · 2025-11-23
> **References**
>
> [1] Rodríguez, Juan, et al. "Electronic nose for quality control of Colombian coffee through the detection of defects in 'Cup Tests'." Sensors 10.1 (2010): 36-46.
>
> [2] Wijaya, Dedy Rahman, Riyanarto Sarno, and Enny Zulaika. "Electronic nose dataset for beef quality monitoring in uncontrolled ambient conditions." Data in Brief 21 (2018): 2414-2420.
>
> [3] Mahradian, Mehrab. "Fruit Freshness, Electronic Nose Dataset." Kaggle dataset (2023).
>
> [4] Fonollosa, Jordi, Irene Rodríguez-Luján, and Ramón Huerta. "Chemical gas sensor array dataset." Data in Brief 3 (2015): 85-89.
>
> [5] Vergara, Alexander. "Gas Sensor Array Drift Dataset at Different Concentrations." UCI Machine Learning Repository (2013).

---

> ### Comment · Reviewer_WutW · 2025-11-27
>
> I thank the authors for the response and for addresing my concerns. I will increase my score

---

> ### Author Response · Authors · 2025-12-03
> **Thanks for raising the score**
>
> We sincerely thank reviewer WutW for actively engaging in our rebuttal and raising the score to 8. We will incorporate all feedbacks in our final manuscript.

---

### Meta-Review · Area_Chair_eCAJ · 2026-01-17

**Summary:**

This paper introduces SmellNet, a large-scale sensor-based olfactory dataset collected with consumer-grade gas sensors, covering 50 single ingredients and 43 controlled ingredient-level mixtures, together with extensive temporal recordings. The paper also proposes ScentFormer, a Transformer-based temporal model, and evaluates a broad set of baselines (MLP, CNN, LSTM, Transformer) across single-ingredient recognition, cross-modal supervision with GC–MS, and mixture composition prediction.
Reviewers broadly appreciated the novelty and potential impact of a hardware-proximal, sensor-based olfactory benchmark, the careful data collection protocol, and the comprehensive experimental evaluation . The main concerns raised across reviews focused on (i) overly strong or potentially misleading positioning claims relative to existing olfactory AI work , (ii) methodological clarity regarding sensor signals, environmental confounds, and temporal generalization , (iii) transparency and interpretation of GC–MS supervision , and (iv) dataset usability issues such as channel reliability and mixture evaluation protocols .
In their rebuttal, the authors provided detailed clarifications, additional ablation studies, and concrete revision plans. These include sensor-only versus environment-only analyses across multiple model families , leave-one-day-out evaluations to assess temporal robustness , revised GC–MS representations based on raw EI spectra and clearer explanations of ingredient-level supervision , additional mixture metrics such as cosine similarity , and clearer documentation of hardware, preprocessing, and LLM usage . The authors also committed to tempering positioning claims and refining the paper’s scope .
Overall, the rebuttal substantially reduced the main risks identified by reviewers. The work is now more clearly framed as a sensor-based, machine-olfaction benchmark, complementary to psychophysical and molecular olfactory datasets. While some limitations remain—particularly regarding robustness to broader environmental variation—these are acknowledged as scope limitations rather than methodological flaws. Reviewer discussion converged toward a generally positive assessment after rebuttal.

**Reviewer Concerns:**

Concerns addressed by the rebuttal:
1.Positioning and over-claiming (Reviewer WutW, Reviewer wU3A): The authors agreed to remove or soften absolute claims and explicitly acknowledge prior large-scale olfactory modeling efforts, narrowing the contribution to sensor-based, time-series olfactory learning with portable hardware.
2.Shortcut learning and environmental confounds (Reviewer uEQe): New ablations demonstrate that environment-only models perform substantially worse than sensor-based models across both neural and tree-based classifiers, mitigating concerns about trivial shortcuts or leakage.
3.Temporal generalization (Reviewer uEQe): Leave-one-day-out evaluations show that models generalize across acquisition days despite measurable domain shifts.
4.GC–MS pairing and representation (Reviewer uEQe, Reviewer wU3A): The authors clarified that GC–MS is used only as ingredient-level label supervision, introduced a standard spectral representation, and showed that conclusions are robust across GC–MS encodings.
5.Mixture evaluation and interpretation (Reviewer uEQe, Reviewer eaQT): Additional metrics and clearer justification of mixture protocols clarify the scope and limitations of mixture prediction.
6.Dataset usability and transparency (Reviewer wU3A): The distinction between the 12-channel hardware and the 6 reliable channels used in experiments is clarified, with a plan to release a clearly documented 6-channel processed dataset and a separately marked raw version. Limited LLM usage is clarified and reproducible artifacts are promised.
Remaining considerations:
Environmental robustness and sensor redundancy (Reviewer uEQe): Generalization to substantially different environments and potential redundancy among correlated sensor channels remain open limitations. These are now explicitly acknowledged by the authors and framed as future work rather than blockers.

**Reviewer Scores:**

After the rebuttal, reviewer sentiment converged positively. Reviewer uEQe and Reviewer WutW explicitly indicated that they would raise their scores following the additional analyses and clarifications. Reviewer eaQT maintained a strong positive recommendation throughout. Reviewer wU3A did not update their score but stated they would not mind acceptance, and their main concerns were addressed through concrete revision plans. Overall, the post-rebuttal reviewer consensus supports acceptance.

---

### Decision · Program_Chairs · 2026-01-26

Accept (Poster)